



# Design, evaluation and future projections of the NARCliM2.0 CORDEX-CMIP6 Australasia regional climate ensemble

Giovanni Di Virgilio[1,2], Jason P. Evans[2,3], Fei Ji[1,3], Eugene Tam[1], Jatin Kala[4], Julia Andrys[4], Christopher Thomas[2], Dipayan Choudhury[1], Carlos Rocha[1], Stephen White[1], Yue Li[1], Moutassem El Rafei[1], Rishav Goyal[1], Matthew L. Riley[1] and Jyothi Lingala[4]

[1]Climate & Atmospheric Science, NSW Department of Climate Change, Energy, the Environment and Water, Sydney, Australia

[2]Climate Change Research Centre, University of New South Wales, Sydney, Australia

[3]Australian Research Council Centre of Excellence for Climate Extremes, University of New South Wales, Sydney, Australia

[4]Environmental and Conservation Sciences, and Centre for Climate Impacted Terrestrial Ecosystems, Harry Butler Institute, Murdoch University, Murdoch, WA 6150, Australia

*Correspondence to*: Giovanni Di Virgilio (giovanni.divirgilio@environment.nsw.gov.au;

giovanni@unsw.edu.au)

**Abstract**. NARCliM2.0 comprises two Weather Research and Forecasting (WRF) regional climate
models (RCMs) downscaling five CMIP6 global climate models contributing to the Coordinated
Regional Downscaling Experiment over Australasia at 20 km resolution, and south-east Australia at 4
km convection-permitting resolution. We first describe NARCliM2.0's design, including selecting
two, definitive RCMs via testing seventy-eight RCMs using different parameterisations for planetary
boundary layer, microphysics, cumulus, radiation, and land surface model (LSM). We then assess
NARCliM2.0's skill in simulating the historical climate versus CMIP3-forced NARCliM1.0 and
CMIP5-forced NARCliM1.5 RCMs and compare differences in future climate projections. RCMs
using the new Noah-MP LSM in WRF with default settings confer substantial improvements in
simulating temperature variables versus RCMs using Noah-Unified. Noah-MP confers smaller
improvements in simulating precipitation, except for large improvements over Australia's southeast
coast. Activating Noah-MP's dynamic vegetation cover and/or runoff options primarily improve
simulation of minimum temperature. NARCliM2.0 confers large reductions in maximum temperature
bias versus NARCliM1.0 and 1.5 (1.x), with small absolute biases of ~0.5K over many regions versus
over ~2K for NARCliM1.x. NARCliM2.0 reduces wet biases versus NARCliM1.x by as much as
50%, but retains dry biases over Australia's north. NARCliM2.0 is biased warmer for minimum
temperature versus NARCliM1.5 which is partly inherited from stronger warm biases in CMIP6





versus CMIP5 GCMs. Under shared socioeconomic pathway (SSP)3-7.0, NARCliM2.0 projects ~3K
warming by 2060-79 over inland regions versus ~2.5K over coastal regions. NARCliM2.0-SSP3-7.0
projects dry futures over most of Australia, except for wet futures over Australia's north and parts of
western Australia which are largest in summer. NARCliM2.0-SSP1-2.6 projects dry changes over
Australia with only few exceptions. NARCliM2.0 is a valuable resource for assessing climate change
impacts on societies and natural systems and informing resilience planning by reducing model biases
versus earlier NARCliM generations and providing more up-to-date future climate projections
utilising CMIP6.

**Keywords:**

Climate change; climate impact adaptation; dynamical downscaling; CORDEX-CMIP6; model
design; model evaluation





## 1. Introduction

Climate projections are foundational to informing climate change mitigation and adaptation planning at various spatial scales (IPCC, 2021). Regional climate models (RCMs) dynamically downscale global climate models (GCMs) at ~100-200 km resolution to simulate higher resolution climate projections that better resolve local-scale influences on regional climate, such as mountain ranges, land-use variation, land-sea contrasts, and convective processes (Torma et al., 2015; Giorgi, 2019). As such, whilst GCMs are the best tools for investigating climate at global scales, RCMs provide improved guidance for climate policy at regional scale, which is the scale at which climate change impacts are experienced (Hsiang et al., 2017).

The NARCliM programme (New South Wales and Australian Regional Climate Modelling) is now in its third generation. Like its predecessors, NARCliM version 2.0 ('NARCliM2.0'), aims to produce robust, detailed regional climate projections at spatial scales relevant for use in local-scale climate change analysis. A key feature of all NARCliM generations is to simulate the climate over the Coordinated Regional Downscaling Experiment (CORDEX)-Australasia domain, and a higher resolution inner domain over southeast Australia via one-way nesting (Figure 1). With one-way nesting the inner domain obtains its initial and lateral boundary conditions from the simulation over CORDEX-Australasia. NARCliM1.0 simulated the climate of Australasia for three periods (1990-2009, 2020-2039, 2060-2079) at 50 km resolution and southeast Australia at 10 km using three configurations of the weather research and forecasting (WRF) RCM (Skamarock et al., 2008) to downscale GCMs from Coupled Model Intercomparison Project phase three (CMIP3) under the SRES A2 greenhouse gas (GHG) scenario (Evans et al., 2014). NARCliM1.5 used CMIP5 GCMs under representative concentration pathways (RCP) 4.5 and 8.5 to simulate continuously for 1950-2100 on the same grids as NARCliM1.0 using two of its RCMs (Nishant et al., 2021).

NARCliM2.0 aims to improve performance in simulating the Australian climate relative to previous NARCliM generations with the goal of better informing community resilience to climate change (New South Wales Government, 2022, 2023). All NARCliM projects include a bottom-up design ethos involving multi-sectoral end-user engagement in specifying model requirements to ensure model performance and outputs meet end-user needs. Key requirements from the NARCliM2.0 user-consultation include providing increased detail in climate simulations via higher resolution, and improving the simulation of precipitation and temperature as these are fundamental inputs to climate impact studies. Whilst NARCliM1.0 and 1.5 (1.x) confer the expected level of performance in simulating the Australian climate (Di Virgilio et al., 2019; Evans et al., 2020b), recent technological and scientific advancements mean that aspects of their performance might now be improved. NARCliM1.x RCMs show widespread cold biases in maximum temperature exceeding −5K for some RCMs. Conversely, minimum temperature is simulated more accurately with biases in the range of

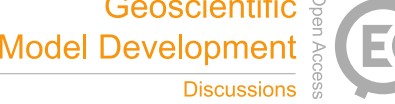

±1.5K. NARCliM1.x RCMs overestimate precipitation, particularly over Australia's socio-
economically important eastern seaboard (Di Virgilio et al., 2019).
As they are expensive to run from both computational and data storage perspectives, dynamical
downscaling projects like NARCliM2.0 use a subset of available GCMs as driving data, necessitating
careful model selection. Similarly, a large combination of different physical parametrisations
available for the WRF RCM enables many structurally different RCMs to be potentially used to
downscale GCMs. A key component of NARCliM2.0's design is testing the viability of alternative
RCM parameterisations via a three-phase approach, with each phase building on the preceding phase
to identify the RCM parameterisations that perform well during testing to meet NARCliM2.0's aim of
improving the simulation of Australia's climate. GCM and RCM statistical independence are also
sought to avoid creating a biased sample of climate change. Hence, the aims of this paper are to:
1) describe how and why NARCliM2.0 differs from its predecessors in terms of its design and
production processes, explaining the model test and evaluation approaches underlying its design
decisions. A key focus is on the design and testing of seventy-eight different WRF RCMs and their
evaluation to identify a subset of RCMs for use in NARCliM2.0;
2) characterise the performance improvements of CMIP6-NARCliM2.0 RCMs in simulating the
Australian climate relative to previous NARCliM generations by evaluating their skill in simulating
mean maximum and minimum temperature and precipitation versus observations;
and 3) summarise the climate projections produced by CMIP6-NARCliM2.0 and how these
differ from previous CMIP3-5-NARCliM generations.
The following section summarises the basic design features of each NARCliM generation;
section 3. describes NARCliM2.0's design process with a focus on its RCM physics testing, as well as
a brief overview of its production process; section 4. describes evaluation methods and metrics;
section 5. summarises the RCM physics test results; section 6. evaluates the performance of all
NARCliM models in simulating the recent Australian climate; section 7. provides an overview of their
future projections; and section 8. discusses key results and summarises this paper.



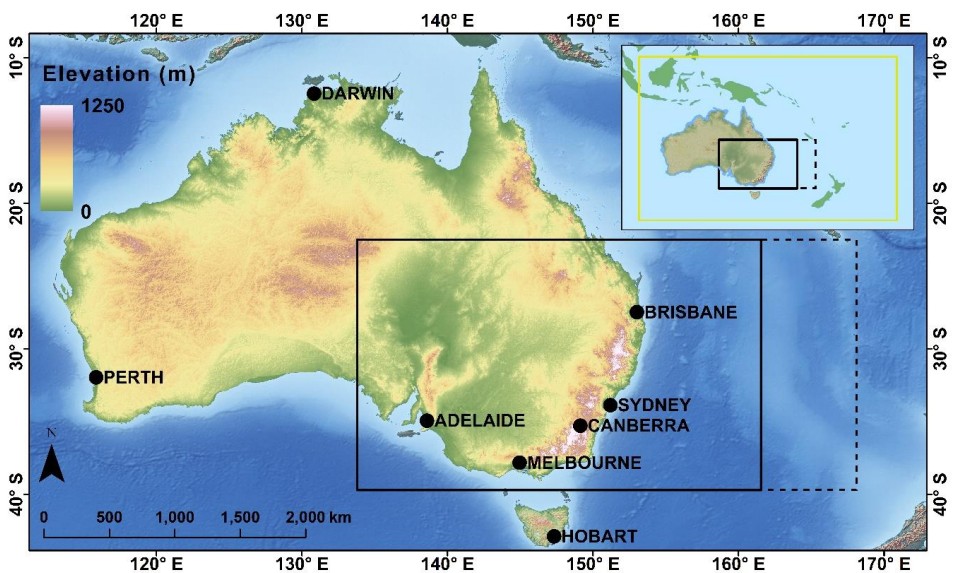


**Figure 1.** Model domains for NARCliM regional climate simulations. The southeast inner domain for
NARCliM2.0 is delineated with a solid black rectangle; the corresponding inner domain for NARCliM1.0 and
1.5 is delineated with a dashed black line. The elevated terrain of the Australian Alps which form part of the
Great Dividing Range is in eastern Australia. Inset shows the CORDEX-Australasia outer domain.

## 2. Three generations of NARCliM: model overviews

The design of NARCliM1.0 is described in Evans et al. (2014); NARCliM1.5 used the same design
approach but used CMIP5 rather than CMIP3 GCMs. All generations of NARCliM use different
versions of the WRF model (Skamarock et al., 2008) to perform dynamical downscaling of GCMs
since the WRF model goes through regular updates. The southeast Australian inner domain captures
five of Australia's eight capital cities (Figure 1) and over 75% of the Australian population
(Australian Bureau Statistics, 2024). Additionally, the inner domain captures coastal regions that are
characterised by topographic complexity and land-use class variation. Regions east of the Great
Dividing Range mountains in southeast Australia (Figure 1) show different responses to oceanic
climate modes compared to inland semi-arid regions (Murphy and Timbal, 2008) and are impacted by
events such as rapidly developing storms, including east coast lows (Pepler and Dowdy, 2021). Such
atmospheric processes are not adequately resolved by GCMs due to coarse resolutions (Di Virgilio et
al., 2022; Grose et al., 2020).

NARCliM2.0 encompasses several design advancements over its predecessors (Table 1).
NARCliM2.0 RCMs have a 20 km resolution CORDEX-Australasia domain (versus 50 km) and 4 km
(versus 10 km) domain over southeast Australia and use 45 (versus 30) vertical levels. The aim of



increasing the resolution of this inner domain from 10 km to 4 km is to render these simulations
convection-permitting (Kendon et al., 2021; Lucas-Picher et al., 2021). Hence, whilst the 20 km-
resolution outer domain uses cumulus parametrisation, simulations over the 4 km domain do not use
cumulus parametrisation. NARCliM2.0 also includes a new collaboration with the Western Australian
government, with separate 4 km simulations being performed over south-west and north-west Western
Australia (not shown in Figure 1) as part of the Western Australian climate science initiative (DWER,
2023). Boundary conditions derived from the 20 km NARCliM2.0 CORDEX Australasia domain are
used to drive these simulations. Additional major differences in model setup for NARCliM2.0
include:
▪ NARCliM1.0 RCMs use different parameterisations for planetary boundary layer (PBL)
physics, surface physics, cumulus physics, land surface model (LSM), and radiation (Evans et
al., 2014). These RCM parameterisations were also used for NARCliM1.5. Owing to the pro-
ject aims stated above, RCM parameterisations for NARCliM2.0 differ to those of NAR-
CliM1.x (see sect. 3).
▪ NARCliM2.0 increases the number of driving GCMs to 5 and simulates for a wider range of
plausible future climates via three shared socioeconomic pathways (SSP). SSP1-2.6 is select-
ed as a low GHG scenario envisaging a future climate with $CO_2$ emissions cut to net zero by
around 2075 and warming held to below 2˚C by 2100; SSP2-4.5 estimates projected warming
under a 'middle of the road' scenario where temperatures increase to ~2.7˚C by 2100; and
SSP3-7.0 is a high GHG scenario which assumes warming of ~4˚C by 2100 (IPCC, 2021).
▪ Urban physics is activated in NARCliM2.0 (WRF setting: sf_urban_physics=1) to represent
surface energy balance in urban areas via a single layer urban canopy model (Kusaka and
Kimura, 2004).
▪ Input of different aerosol species is activated for the RCM radiation scheme using the Tegen
et al. (1997) climatology available in WRF (aer_opt=1). This aerosol forcing is the same for
all GCMs, and not model-specific.
▪ The eastern boundary of the NARCliM2.0 inner domain is located further westward relative
to that of NARCliM1.x (Figure 1).





**Table 1**. High-level design features of three generations of NARCliM regional climate models

| | Model Generation | | |
|---|---|---|---|
| | NARCliM1.0 | NARCliM1.5 | NARCliM2.0 |
| **Release date** | 2014 | 2020 | 2023-2024 |
| **Years simulated** | 1990-2009, 2020-2039, 2060-2079 | 1950-2100 | 1950-2100 |
| **Grid resolutions: CORDEX-Australasia; NARCliM inner domains** | 50 km; 10 km | 50 km; 10 km | 20 km; 4 km |
| **Vertical levels** | 30 | 30 | 45 |
| **Global Climate Models** | 4 CMIP3 GCMs | 3 CMIP5 GCMs | 5 CMIP6 GCMs |
| **Regional Climate Models** | 3 RCM configurations (WRF3.3) | 2 RCM configurations (WRF3.6.0.5) | 2 RCM configurations (WRF4.1.2) |
| **Future emission scenarios** | SRES A2 | RCP4.5, RCP8.5 | SSP1-2.6, SSP2-4.5, SSP3-7.0 |
| **Reanalysis-driven (CORDEX Evaluation)** | NCEP: 1950-2009 | ERA-Interim: 1979-2013 | ERA5: 1979-2020 |

## 3. NARCliM2.0 design and production process overview

The NARCliM2.0 design and production processes are summarised below in reference to Figure 2.

The design process is an adaptation of that introduced in Evans et al. (2014). Two companion

manuscripts describe elements shown in Figure 2, and which are therefore only summarised briefly in

this manuscript. Di Virgilio et al. (2022) describes the CMIP6 GCM selection process summarised in

Box 2, and Di Virgilio et al. (in review) describes the ERA5 evaluation undertaken in Boxes 5 and 6.

**I. Design Phase:**

i)  **Box 1:** model design requirements are identified via consultation between NARCliM2.0

modelling groups and multi-sectoral end-users, as well as adherence to CORDEX-CMIP6

design requirements (WCRP, 2020).





ii)    **Box 2:** NARCliM1.x selected driving CMIP3-5 GCMs (respectively) via literature review
of existing GCM evaluations. During NARCliM2.0 design, there were no pre-existing
comprehensive evaluations of individual CMIP6 GCMs for the Australian region, includ-
ing assessments of climate change signals and GCM statistical independence. Hence, an
evaluation and selection of CMIP6 GCMs was conducted (see Di Virgilio et al. 2022).
This evaluation selected five GCMs to force two NARCliM2.0 RCMs (see sect 3.2 and
3.4). The relative contribution to uncertainty/variation in climate projections can be larger
for GCMs than for RCMs (e.g. Lee et al., 2023).
iii)    **Box 3:** a new WRF RCM multi-physics test ensemble is created for NARCliM2.0: RCM
physics testing is conducted via a three-phase approach, with each phase building on the
findings of the preceding phase to identify the RCM parameterisations that perform well
during testing with the aim of improving the simulation of the Australian climate. In this
way, RCMs are parameterised with different physics settings via each test phase, system-
atically removing poor performing options while facilitating the fine tuning and im-
provement of the parameterisations that perform well during testing to build a total en-
semble size of seventy-eight structurally different test RCMs. The performances of the
different test RCM configurations are evaluated, ultimately selecting a subset of seven
RCMs for subsequent downscaling of ERA5 reanalysis and comprising the CORDEX
evaluation experiment.
iv)    **Boxes 4-6:** These seven RCMs are used to downscale ERA5 reanalysis over the 20 km
and 4 km domains for 1979-2020. Evaluating these ERA5-forced simulations informs se-
lection of two 'production' RCMs for CMIP6-forced downscaling (see sect. 3.4 and Di
Virgilio et al. in review).
**II. Production Phase:**
i)    **Boxes 7-8:** CMIP6 GCM data are pre-processed to create initial and boundary conditions
to drive simulations for the historical (1950-2014) and SSP experiments (2015-2100). A
code repository used for this GCM preprocessing is available at
https://bitbucket.org/oehcas/narclim2-
0_design_and_evaluation_2024_support_materials/src/main/ within the
WRF/repo_snapshots subdirectory. Quality assurance/quality control (QA/QC) is per-
formed on these data before initiating the simulations (e.g. variables are checked to con-
firm data do not contain significant outliers across ensemble members).
ii)    **Boxes 9-11:** the 151-year CMIP6-forced NARCliM2.0 RCM simulations are run using
National Computing Infrastructure at Canberra, Australia (NCI, https://nci.org.au/). File
integrity verification and QA/QC are performed on each year of raw WRF output
throughout the simulation lifecycle and prior to post-processing to CORDEX-compliant



format climate variables. QA/QC tests include calculating the minimum, maximum, mean and standard deviation for key variables over consecutive periods of six days. Variables are categorised as either normally distributed or otherwise. Normally distributed variables (e.g. surface temperature) are deemed potentially erroneous if their minima/maxima are greater than five standard deviations away from the global mean of the relevant statistic of the rolling six-day period. Non-normally distributed variables (e.g. snow depth and precipitation) are checked for global minima and maxima only.

iii) **Boxes 12-13:** after each year of simulation raw output is generated, their post-processing is initiated to produce CORDEX CORE, Tier 1 and Tier 2 variables (WCRP, 2022). A statistical QA/QC process is automatically applied to each year of post-processed CORDEX CORE variables as they are generated throughout the simulations. QA/QC tests include:

- Check for presence of missing values.
- Check that all values are within realistic ranges for minima and maxima.
- Check minima and maxima are not equal at any timestep with exceptions (e.g. snow depth which can be zero everywhere in the outer domain).
- Check that changes over time are within realistic ranges (i.e. assess temporal gradients).
- Check that changes between neighbouring data points are within realistic ranges (i.e. assess spatial gradients).
- Check the number of grid cells with NaN (non-numerical) values do not exceed the threshold set for the variable.

Reasonable ranges for variables are determined using a series of threshold values that are based on historical records and/or empirical analysis. QA/QC computer scripts generate 'exceedance files' which output every data point that surpasses the threshold values, and these exceedance files are then manually reviewed to determine whether an issue is a true or false positive, etc.

iv) **Box 14:** Once each year of WRF raw files are post-processed, raw files are transferred to a tape facility for long-term storage.



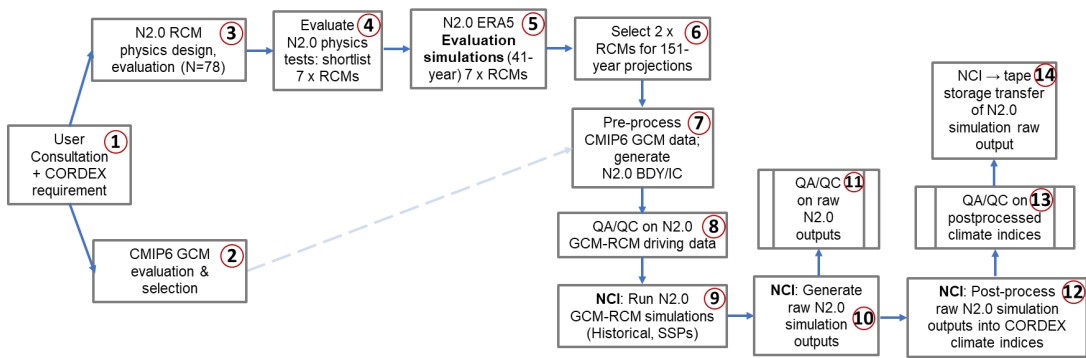

**Figure 2.** Simplified overview of NARCliM2.0 (N2.0) design and production processes. ERA5 = ECMWF Reanalysis v5 data; BDY = boundary conditions; IC = Initial conditions; QA/QC = Quality Assurance / Quality Control; NCI = National Computing Infrastructure (high performance computer used for N2.0 production simulations).

These model design and production stages are now described in more detail:

## 3.1 Model evaluation and selection

Practical constraints such as available compute and data storage resources enforce an upper limit on GCM-RCM ensemble size. Thus, NARCliM2.0 uses a subset of available CMIP6 GCMs and WRF RCM configurations, necessitating careful GCM and RCM selection to create a subset of GCM-RCMs that provide robust climate simulations whilst also adequately sampling model uncertainty. In selecting a subset of GCMs and RCMs for dynamical downscaling, it is desirable to reject models that perform consistently poorly relative to their peers in simulating the current climate, as this provides lower confidence in the projected change (Evans et al., 2020b; Di Virgilio et al., 2022; Grose et al., 2023). Furthermore, the modelled climate space sampled is reduced when selecting a subset of GCMs, which can create a biased view of the climate, as well as the plausible change in climate. Care must therefore be taken to ensure that the subset of models used for downscaling are representative of the full range of possible climates, and that model errors are uncorrelated, i.e., that models are statistically independent. The steps taken to evaluate and select GCMs and RCMs for NARCliM2.0 are described next.

## 3.2 CMIP6 GCM evaluation

A three-phase process was used to evaluate individual CMIP6 GCMs (for further details see Di Virgilio et al. 2022):


### 3.2.1 CMIP6 GCM Performance

The performances of individual CMIP6 GCMs in simulating the Australian climate were assessed with respect to climate means, extremes, climate modes, and daily climate variable distributions. A set of GCMs that performed consistently poorly across the variables and statistics considered were identified. These models, as well as those with insufficient data to enable dynamical downscaling using the WRF RCM, were excluded from further evaluation leaving 27 GCMs for subsequent assessment.

### 3.2.2 CMIP6 GCM Independence

The retained 27 GCMs were subjected to the Bishop and Abramowitz (2013) and Herger et al. (2018) independence analyses (see sect. 4.4). The GCMs were then ranked according to their relative level of statistical independence.

### 3.2.3 Sampling CMIP6 GCM Climate Change Spread

For climate change risk assessments, climate projections should reflect as much of the range of plausible future climate changes as possible (Whetton and Hennessy, 2010). The subset of CMIP6 GCMs selected for NARCliM2.0 spanned a wide range of future changes in annual mean temperature and precipitation. Climate change signals were calculated for 2080-2099 minus 1995-2014 for the Australian continent and south-east Australia under SSP3-7.0 (for the latter, see Figure 3). The GCM independence rankings were placed within this climate change space, with higher independence rankings viewed as favourable, along with consideration of the following criteria:

i) A balanced range of GCM Equilibrium Climate Sensitivities (ECS) were sampled. ECS is the long-term increase in global mean surface air temperature in response to the radiative forcing caused by a doubling of pre-industrial $CO_2$ concentrations. ECS is related to global temperature change, not just changes over Australia, however, it correlates strongly with regional warming. Around one third of CMIP6 GCMs show ECS values higher than the upper end of the likely range of 2.5°C to 4°C (IPCC, 2021). An upper range of $> \sim 5$°C cannot be ruled out (Meehl et al., 2020; Bjordal et al., 2020; Sherwood et al., 2020).

ii) Some CMIP6 GCMs that are favourable in terms of model performance and independence could not be selected as input to WRF for NARCliM2.0 owing to insufficient data availability for key variables/variable, where ideally, WRF requires sub-daily data for the variables shown in Supporting Information, Table S1.

As a result of the above process, the five CMIP6 GCMs listed in Table 2 are selected to force NARCliM2.0 RCMs.



**Table 2.** Basic details of the CMIP6 GCMs used for NARCliM2.0 simulations.

| CMIP6 GCM | Institution | Variant/Run | Atmosphere lat/lon grid (º) |
|---|---|---|---|
| ACCESS-ESM1-5 | CSIRO | r6i1p1f1 | $1.2 \times 1.8$ |
| EC-Earth3-Veg | EC-EARTH consortium | r1i1p1f1 | $0.7 \times 0.7$ |
| MPI-ESM1-2-HR | Max Planck Institute for Meteorology (MPI) | r1i1p1f1 | ~0.9 |
| NorESM2-MM | Norwegian Climate Centre | r1i1p1f1 | $0.9 \times 0.9$ |
| UKESM1-0-LL | UK Met Office and NERC research centres | r1i1p1f2 | $1.3 \times 1.9$ |

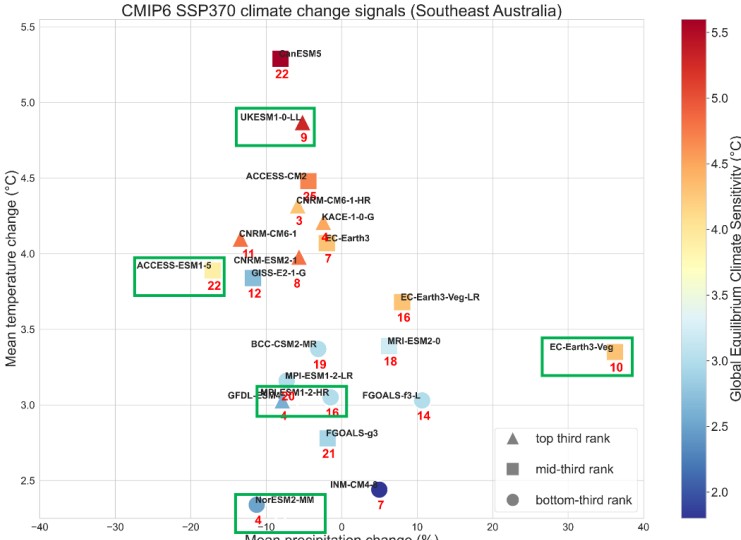

**Figure 3.** CMIP6 GCM climate change signals (2080-2099 versus 1995-2014) over south-east Australia for the
subset of GCMs retained following the model performance evaluation in Di Virgilio et al. (2022), and that
simulated at least monthly mean near surface air temperature and precipitation for the SSP-3.70 scenario. Boxed
GCMs are selected to force NARCliM2.0 RCMs. Marker shapes indicate overall GCM performance; markers
are coloured according to their global equilibrium climate sensitivity (ECS) values; **Red** numbers represent the
smallest Herger Method 1 set for that GCM.

## 3.3 NARCliM2.0 RCM physics testing

The NARCliM2.0 RCM physics testing aims to identify and exclude RCMs that perform consistently
poorly in simulating the southeast Australian climate and to select RCMs that have high statistical
independence. The selection of RCMs in NARCliM2.0 involves the creation of a multi-physics
ensemble where each RCM uses different physical parametrisations for PBL, microphysics, cumulus,
radiation, and LSM. This enables many structurally different RCMs to be constructed and tested. In



NARCliM1.0, 36 WRF RCM configurations were designed, tested, and evaluated (Evans et al. 2014).
NARCliM2.0 physics testing assesses 78 RCM configurations which are progressively tested via three
phases, where each test phase is informed by the outcomes of the preceding phase to systematically
remove poor performing RCM options while facilitating the selection of parameterisations that
perform well during testing. The N=36 RCMs tested for NARCliM1.0 were evaluated based on eight
representative storm event simulations each of two-weeks duration (Evans et al. 2014). NARCliM2.0
physics simulations were run over an entire annual cycle (2016) with a two-month spin up period
commencing 1 November 2015. Australia experienced a range of weather extremes during 2016
driven by a range of climatic influences making 2016 a suitable target year (Bureau of Meteorology,
2017). Whilst assessing RCMs for an entire year improves on assessing for discrete storm events as
per physics testing for NARCliM1.0, it was not feasible to run a large RCM physics ensemble for a
longer duration. Initial and boundary conditions for all phases of the NARCliM2.0 RCM physics test
simulations were derived from the ERA-Interim reanalysis data set (Dee et al., 2011). ERA-Interim
was used because ERA5 was not available at the time. The three phases of NARCliM2.0 physics
testing are as follows:

### 3.3.1 Phase I (N=36)

Thirty-six RCMs were evaluated in Phase I. One radiation scheme (RRTMG) is tested for both long
and short-wave radiation (it is held fixed for all RCMs), whereas physics settings for PBL,
microphysics, cumulus, and LSM are varied. Of the 36 simulations, 18 used the Noah-Unified LSM,
whilst the remainder used Community Land Model version 4.0 (CLM4). The physics options tested
are listed in Table 3, where these were selected based on literature review. Each physics test
simulation is denoted by a 12-digit identifier which comprises 6 pairs of digits, with each pair
corresponding to the choice of a specific physics option as specified in the WRF namelist.input file.
These pairs of digits follow the order: planetary boundary layer (pbl) ¦ cloud microphysics (mp) ¦
cumulus convection (cu) ¦ shortwave radiation (sw) ¦ longwave radiation (lw) ¦ LSM (sf) and
correspond to the WRF namelist options shown in Table 3. For example, the simulation
'050601040402' is interpreted as: 05 ¦ 06 ¦ 01 ¦ 04 ¦ 04 ¦ 02 and denotes that this simulation uses the
following physics settings:

| | |
|---|---|
| bl_pbl_physics | = 05 (MYNN2) |
| mp_physics | = 06 (WSM6) |
| cu_physics | = 01 (Kain-Fritsch) |
| ra_sw_physics | = 04 (RRTMG) |
| ra_lw_physics | = 04 (RRTMG) |
| sf_surface_physics | = 02 (Noah Unified) |





The complete set of WRF RCM configurations tested in Phase I is shown in Supporting Information
Table S2.
**Table 3.** Physics options used in phase I (N=36) tests.

| Physics Option Description | WRF Namelist | Options Tested |
|---|---|---|
| Planetary boundary layer | bl_pbl_physics | 01 = YSU |
| | | 05 = MYNN2 |
| | | 07 = ACM2 |
| Microphysics | mp_physcis | 06 = WSM6 |
| | | 08 = Thompson |
| Cumulus parameterisation | cu_physics | 01 = Kain-Fritsch |
| | | 02 = BMJ |
| | | 06 = Tiedtke |
| Shortwave radiation | ra_sw_physics | 04 = RRTMG |
| Longwave radiation | ra_lw_physics | 04 = RRTMG |
| Land surface model | sf_surface_physics | 02 = Noah-Unified |
| | | 05 = Community Land Model V4 |

*3.3.2 Phase II (N=60): additional LSM and radiation scheme tests*
Phase I RCMs using CLM4.0 were omitted from further testing because they did not consistently im-
prove performance in simulating the Australian climate relative to RCMs using Noah-Unified. In ad-
dition, RCMs using CLM4.0 had increased simulation times (by approximately twice when compared
to Noah-Unified). Hence, Phase II focuses exclusively on further testing of the RCM configurations
that used the Noah-Unified LSM.
The physics settings tested in Phase II are an alternative LSM to Noah-Unified (Noah Multi-
Parameterisation; 'Noah-MP', Niu et al., 2011) and New Goddard radiation. Owing to time/resource
constraints, testing all eighteen Phase I RCMs using Noah-Unified was not feasible. To reduce the
number of RCMs for further testing, the worst-performing Noah-Unified based RCM configurations
identified in Phase I were excluded. The N=18 RCMs using Noah-Unified are listed along with their
overall performance total scores in Table 4 where the lowest scores under 'Rank totals' indicate the
RCMs that overall perform relatively well versus their peers (see sect. 4 Evaluation Methods). Note
that the 'Overall rank' denotes the RCMs' relative ranking among all Phase I RCMs. There is a sharp
reduction in rank totals for RCMs #13-18 inclusive, relative to RCMs #1-12. Therefore, RCMs #13-
18 are excluded from further testing, and RCMs #1-12 are retained.
**Table 4.** RCM physics combination ranks of the Phase I, N=18 Noah Unified (NU) based RCMs.
Scores/ranks are based on model bias and root mean square error for annual and seasonal precipita-
tion, minimum temperature, maximum temperature, climate extremes (wettest and hottest days), and
Perkins Skill Scores (see sect. 4). RCMs #1-12 are selected for further testing.



| RCM # | RCM ID | Physics combination | | | | | Rank total | Overall rank in N=36 Phase I |
|---|---|---|---|---|---|---|---|---|
| | | PBL | MP | Cumulus | SW/LW | LSM | | |
| 1 | 070801040402 | ACM2 | Thom | KF | RRTMG | NU | 484 | 1 |
| 2 | 070601040402 | ACM3 | WSM6 | KF | RRTMG | NU | 495 | 2 |
| 3 | 070802040402 | ACM4 | Thom | BMJ | RRTMG | NU | 527 | 3 |
| 4 | 070602040402 | ACM5 | WSM6 | BMJ | RRTMG | NU | 559 | 4 |
| 5 | 010802040402 | YSU | Thom | BMJ | RRTMG | NU | 574 | 7 |
| 6 | 050801040402 | MYNN2 | Thom | KF | RRTMG | NU | 583 | 8 |
| 7 | 010801040402 | YSU | Thompson | KF | RRTMG | NU | 617 | 11 |
| 8 | 050802040402 | MYNN2 | Thompson | BMJ | RRTMG | NU | 630 | 12 |
| 9 | 070606040402 | ACM2 | WSM6 | Tiedtke | RRTMG | NU | 639 | 13 |
| 10 | 050601040402 | MYNN2 | WSM6 | KF | RRTMG | NU | 662 | 16 |
| 11 | 070806040402 | ACM2 | Thompson | Tiedtke | RRTMG | NU | 662 | 16 |
| 12 | 010602040402 | YSU | WSM6 | BMJ | RRTMG | NU | 674 | 19 |
| 13 | 010601040402 | YSU | WSM6 | KF | RRTMG | NU | 702 | 23 |
| 14 | 010606040402 | YSU | WSM6 | Tiedtke | RRTMG | NU | 759 | 25 |
| 15 | 050606040402 | MYNN2 | WSM6 | Tiedtke | RRTMG | NU | 766 | 27 |
| 16 | 050602040402 | MYNN2 | WSM6 | BMJ | RRTMG | NU | 811 | 31 |
| 17 | 010806040402 | YSU | Thompson | Tiedtke | RRTMG | NU | 830 | 34 |
| 18 | 050806040402 | MYNN2 | Thompson | Tiedtke | RRTMG | NU | 857 | 35 |

This gives two sets of physics combinations for additional testing: 1) one replaces only RRTMG
(|04|04|) for short and longwave radiation with New Goddard (|05|05|) making no other changes; and
2) RRTMG radiation is retained, but Noah-MP (|04|) replaces Noah-Unified (|02|). This creates an
additional 24 RCM configurations for assessment, bringing the total RCMs tested to 60. Although
Noah-MP has several parameter options, Phase II uses its default settings.

### 3.3.3 Phase III (N=78): parameterising Noah-MP

Phase II shows that RCM performance using New Goddard radiation is generally inferior to the same
RCMs using RRTMG (see sect. 5. RCM Physics test results). Consequently, RRTMG radiation is re-
adopted for Phase III. Conversely, a general performance improvement is conferred by using Noah-
MP over Noah-Unified (sect. 5). Given this performance improvement using Noah-MP with default
settings, Phase III assesses RCM performances using specific parameter settings for Noah-MP.
Noah-MP provides a 'dynamic vegetation cover' model option (referred to as dynamic vege-
tation in the WRF users' guide) (Niu et al., 2011). When deactivated (the default), monthly leaf area
index (LAI) is prescribed for various vegetation types and the greenness vegetation fraction (GVF)
comes from monthly GVF climatological values. Conversely, when dynamic vegetation cover is acti-





vated, LAI and GVF are calculated using a dynamic leaf model. We clarify here that dominant plant-
functional types do not change when using this option, but only the LAI and GVF, i.e. only the
amount of green cover changes.
Noah-MP also provides several options for modelling surface run-off and groundwater pro-
cesses including a TOPMODEL (TOPography based hydrological MODEL)-based surface runoff
scheme and a simple groundwater model (SIMGM; Niu et al., 2011). Some studies have shown using
this option improves modelling of soil moisture (e.g. Zhuo et al., 2019). Thus, three new sets of phys-
ics configurations are tested using Noah-MP where default options for specific settings are changed as
follows:
1. activate dynamic vegetation cover (dveg=2 in the WRF namelist); no other changes.
2. activate TOPMODEL runoff with simple groundwater (opt_run=1); no other changes.
3. activate both dynamic vegetation and TOPMODEL runoff with simple groundwater, no other
changes.
As above, the worst performing RCMs in Phase II are excluded from Phase III testing. Based
on the RCM configuration performance rankings (Table 5), there is a sharp reduction in performance
starting from RCM #7 inclusive. Therefore, RCMs #7-12 are excluded from further testing. Phase III
thus comprises 18 new test simulations (sets 1-3 each comprising 6 RCMs) bringing the total RCMs
tested to N=78. Phase III physics tests are denoted using the same RCM identification schemes distin-
guished by appending 'set_1', 'set_2', 'set_3' to identifiers.
**Table 5.** RCM physics combination ranks of the Phase II Noah-MP RCMs. Scores/ranks are based on model
bias and root mean square error for annual and seasonal precipitation, minimum temperature, maximum temper-
ature, climate extremes (wettest and hottest days), and Perkins Skill Scores (see sect. 4).

| No. | Physics combination | Rank total |
|-----|---------------------|------------|
| 1 | 50801040404 | 721 |
| 2 | 70806040404 | 822 |
| 3 | 50802040404 | 848 |
| 4 | 70802040404 | 872 |
| 5 | 70601040404 | 880 |
| 6 | 50601040404 | 891 |
| 7 | 10802040404 | 988 |
| 8 | 70602040404 | 1005 |
| 9 | 70606040404 | 1028 |
| 10 | 10801040404 | 1042 |
| 11 | 70801040404 | 1056 |
| 12 | 10602040404 | 1264 |





*3.3.4 Shortlisting Physics Test RCMs for ERA5-NARCliM2.0 evaluation simulations*
Considering the complete NARCliM2.0 N=78 physics test ensemble, to identify physics test RCMs
that perform poorly overall, RCMs are eliminated if they are in the lowest 1/3 for RCM performance
ranks for any of maximum temperature, minimum temperature, precipitation, or for the overall model
performance rank across these variables (see sect. 5. RCM Physics test results). Under this scheme, 20
RCMs remain. The independence measures are then applied to the remaining 20 RCMs to choose a
final subset of 7 RCMs for ERA5-forced evaluation simulations (see sect. 3.4). The ensemble size
limit of N=7 is determined by available compute resources. These 7 candidate RCMs are assessed for
potential use in the CMIP6 GCM-forced downscaling phase of NARCliM2.0 (sect. 3.4 and Di Virgil-
io et al. in review).

## 381     3.4 CORDEX ERA5-NARCliM2.0 evaluation simulations

NARCliM1.x performed production climate simulations using a two-phase process. Its RCM physics
testing selected definitive 'production-grade' RCMs which were then used to downscale both reanaly-
sis data and CMIP3/5 GCMs. In contrast, for NARCliM2.0, as described above the N=78 RCM phys-
ics testing culminates in shortlisting 7 'production-candidate' RCMs which are used to downscale the
ERA5 reanalysis for 42-years (1979-2020). This enables assessment of shortlisted RCM performances
over a climatological period rather than the single year (2016) of the physics testing, which helps as-
certain that performance differences between shortlisted RCMs are robust across a multi-decadal
timescale capturing climatologically diverse years. The aim is that two definitive production-grade
RCMs can be selected for CMIP6-forced downscaling from these ERA5-forced CORDEX 'evalua-
tion' simulations. Thus, the seven ERA5-NARCliM2.0 RCMs were driven by ERA5.0 boundary con-
ditions for January 1979 to December 2020 using the model and nested domain setups described
above for NARCliM2.0. The skill of these RCMs in simulating the recent Australian climate was as-
sessed as follows (see Di Virgilio et al. in review): annual and seasonal means were calculated for
maximum and minimum temperature and precipitation using monthly means for temperature varia-
bles, and the monthly sum for precipitation. Extremes of maximum temperature and precipitation (99th
percentiles) and extreme minimum temperature (1st percentile) were calculated using daily data. RCM
performances in reproducing observations over these timescales were assessed by calculating model
outputs minus observations (i.e. model bias), and the RMSE of modelled versus observed fields. RCM
skill in simulating distributions of observed variables was assessed by comparing the probability den-
sity functions (PDFs) for daily mean observations versus those of the RCMs. The ultimate outcome of
these ERA5-forced simulations and their evaluation is the selection of two RCM configurations, R3
and R5 to run the CMIP6-forced phase of NARCliM2.0, see Di Virgilio et al. (in review) for further
details on the evaluation methods and results. Supporting Information Figure S1 shows the WRF
namelist settings for the R3 and R5 RCMs (see also sect. 9. Code Availability).



## 3.5 CORDEX CMIP6-forced NARCliM2.0 simulations

The ideal CMIP6 GCM variables and their frequencies required to run the WRF RCM are listed in Table S1. A minority of variables in Table S1 are not available at sub-daily frequencies for every target GCM. This necessitates assumptions/data proxies to be made. For instance, soil moisture and soil temperature variables were unavailable for some selected GCMs; hence, surrogate data, such as surface temperature, were used for initialisation (noting that soil data are only used by the RCM at initialisation). In these cases, we investigated how long it took for uncertainty in the initial conditions to disappear from the WRF output by analysing the regionally averaged soil moisture time series. The data were regionalised according to the four Australian Natural Resource Management (NRM) regions / climate zones (Supporting Information Figure S2) which are broadly aligned with climatological boundaries (Fiddes et al., 2021) and with the IPCC reference regions (Iturbide et al., 2020). Time series plots (Figure S3) show that soil moisture equilibrates to be within a normal range following initialisation, indicating that the 12-month spin-up year (1950) is sufficient to account for the assumptions made at model initialisation.

Boundary and initial conditions were prepared using selected GCM data to run the 151-year GCM-driven simulations using WRF version 4.1.2. The GCM-driven simulations were run and completed using the pre-defined RCM settings for two RCM configurations using the WRF namelists in Supporting Information Figure S1 (see also sect. 9. Code Availability). A cold restart was performed on the last Historical experiment year (2014), thus enabling the SSP1-2.6 and SSP3-7.0 experiments to be run for 2015-2100 concurrently with the Historical experiment. The 2014 cold start year is eventually overwritten by Historical runs initiated in 1950.

# 4. Evaluation methods

This section largely focuses on the methods and metrics used for the NARCliM2.0 RCM physics testing. Overviews of the methods and metrics for CMIP6 GCM evaluation and selection and assessments of the ERA5-forced evaluation simulations are provided above, with further information on these available in Di Virgilio et al. (2022) and Di Virgilio et al. (in review).

## 4.1 Observations

Australian Gridded Climate Data (AGCD version 1.0; Evans et al., 2020a) are the observational data used to evaluate the NARCliM2.0 RCM physics test RCMs. These daily gridded data for maximum and minimum temperature and precipitation are obtained from an interpolation of station observations across Australia. AGCD data are on a regular WGS84 grid with a grid-averaged resolution of 0.05°. For the NARCliM2.0 RCM physics tests, the AGCD data were re-gridded to correspond with the RCM data from the inner domain on their native grids using a conservative area-weighted re-gridding





scheme. All data (RCM and AGCD) were restricted to a common extent contained within the inner domain over southeast Australia, and a land mask was applied so that statistics were computed using only land pixels. Treatment of AGCD for the CMIP6 GCM evaluation and the ERA5-NARCliM2.0 RCM evaluation is described in Di Virgilio et al. (2022) and Di Virgilio et al. (in review), respectively.

## 4.2 Methods and metrics: phase I-III physics tests

RCM performances in reproducing observations for daily maximum and minimum temperature and daily precipitation were assessed by calculating the model bias, i.e. model outputs minus AGCD, and the RMSE of modelled versus observed fields. Model biases and RMSEs were calculated at annual and seasonal timescales. The model representations of the hottest and the wettest day on an annual time scale over the study region were also compared with AGCD. PDFs were calculated for each variable using daily data. The Perkins skill score (PSS) (Perkins et al., 2007) was calculated to assess the overall degree of overlap between modelled and observed distributions, with PSS = 1 indicating that distributions overlap perfectly.

To identify the overall performances of the RCM configurations, the RCMs are ranked based on the bias and RMSE for all variables and seasons, the annual PSS, as well as the bias and RMSEs for the maximum temperature and precipitation extremes. These ranks are then summed with the lowest totals indicating the best performing RCM configurations overall.

## 4.3 CMIP6 GCM and ERA5-NARCliM2.0 evaluations

Overviews of the evaluation methods and rationale for these components of NARCliM2.0 design have been provided above. For further details on methods and results on the CMIP6 GCM evaluation and the ERA5-NARCliM2.0 RCM evaluation, see Di Virgilio et al. (2022) and Di Virgilio et al. (in review), respectively.

## 4.4 Independence assessments

We used the method of Bishop and Abramowitz (2013) as one of two methods of assessing the independence of physics test RCMs and the target CMIP6 GCMs under evaluation for use in NARCliM2.0. This approach uses the covariance in model errors as the basis to define model dependence; specifically, independence coefficients are derived from the error covariance matrix of the RCMs or GCMs. Model independence is quantified using the correlation of model errors. For the physics test RCMs, errors are computed by comparing the climatology of maximum and minimum temperature and precipitation over the south-east Australia inner domain for 2016 with corresponding AGCD observations. The same calculation is performed for the CMIP6 GCMs, except for the Australian continent. Daily timeseries of precipitation, maximum and minimum temperature are calculated individual-





ly for each RCM and for AGCD. The simulated and observed daily timeseries of each variable are
then normalised by the standard deviation of the corresponding observed variable. These normalised
variables are concatenated for each RCM (GCM) and AGCD. An anomaly time series for each grid
cell is then produced. These time series are used to create a 'model error covariance matrix' contain-
ing the errors for all RCMs (GCMs). The coefficients of a linear combination of the RCMs (GCMs)
that optimally minimises the mean square error depends on both model performance and model de-
pendence (Bishop and Abramowitz, 2013). The result of this minimisation problem is written in terms
of the covariance matrix. The magnitude of coefficients assigned to each RCM (GCM) reflects a
combination of their performance and independence. Highly independent models have different errors
when simulating the recent climate. Models with the largest coefficients have the most independent
errors versus observations.
The Herger method of subset selection (Herger et al., 2018), as implemented here, uses quad-
ratic integer programming to find the subset of models whose equally-weighted subset mean (EWSM)
minimises a quadratic cost function. This cost function is chosen to measure the performance of the
EWSM in comparison to a given observational product. The two cost functions used here are: the
mean squared error (MSE) between the EWSM and the observational product (Herger et al. 2018, Eq.
1); and another which measures a combination of the MSE of the EWSM, the average MSE of each
subset member, and the average pairwise mean squared distance between subset members (Herger et
al. 2018, Eq. 2).

## 491 4.5 NARCliM2 CMIP6-RCMs: historical evaluation and climate change
## 492 projections

Performances of NARCliM2.0 versus NARCliM1.x RCMs in reproducing the recent Australian cli-
mate are evaluated by calculating the model biases (model outputs minus AGCD observations) for
mean maximum and minimum temperature and precipitation for 1990-2009. To enable comparison of
future projections between NARCliM1.0, NARCliM1.5 and NARCliM2.0 (where NARCliM1.0 mod-
elled for 1990-2009, 2020-2039, and 2060-2079), all NARCliM ensemble projected changes are
shown as far future (2060–2079) minus present day (1990–2009).

### 499 4.6 Statistical significance

When quantifying future climate change projections (compared to the historical period) and biases in
maximum and minimum temperature, the statistical significance is calculated for each grid cell using
t-tests ($\alpha = 0.05$) assuming equal variance. The Mann–Whitney U test is used for precipitation given
its non-normality. For individual RCMs, grid cells showing statistically significant changes are stip-
pled, otherwise they are shown in colour where change is statistically insignificant. Results on the
statistical significance of each ensemble mean are separated into three categories following Tebaldi et

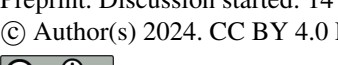



al. (2011): 1) statistically insignificant areas are shown in colour, denoting that less than 50% of
RCMs are significantly biased/different; 2) in areas of significant agreement (stippled), at least 50%
of RCMs are significantly biased/different and at least 70% of significant models in the CMIP6-
NARCliM2.0 RCM ensemble agree on the sign of the bias/difference. In such areas, many ensemble
members have the same bias sign which is an undesirable outcome; and 3) areas of significant disa-
greement, where at least 50% of RCMs are significantly biased/different and fewer than 70% of sig-
nificant models agree on the bias sign, are shown with diagonal hatching for the CMIP6-NARCliM2.0
historical evaluation and climate change signals.

## 5. RCM Physics test results

## 5.1 Phase I RCM performance summary

The spatial variation and magnitudes for Phase I RCM biases and RMSEs for annual mean maximum
and minimum temperature and precipitation are shown in Figures 4-5, respectively. Overall, RCMs
are biased cold for maximum temperature (mean absolute bias for the ensemble mean = 1.18 K), and
warm-biased for minimum temperature (mean absolute bias = 1.31 K; Figure 4a-b). Maximum tem-
perature RMSE magnitudes are large over the elevated terrain of the southeast coast and over western
regions (Figure 5a). The simulation of precipitation shows biases of varying sign, with wet biases that
are strongest over eastern coastal regions (Figure 4c). Precipitation RMSEs are particularly large
along the eastern coastline (>15 mm), and generally show an east-west gradient, i.e. progressively
decreasing further inland from the coast (Figure 5c).

## 5.2 Comparing Phase II Physics Test RCM performances versus Phase I

### 5.2.1 Climate Means

Overall, the RCM ensemble using New Goddard (NG) radiation has inferior performance to the corre-
sponding RCMs using RRTMG in terms of annual/seasonal mean maximum temperature biases,
RMSEs, and PSS (Table 7). In contrast, NG confers superior performance for annual/seasonal mean
minimum temperature for these statistics. RCMs using NG show reduced biases for annual mean and
spring-time precipitation, but larger errors for DJF and JJA (Table 7). RMSEs for annual and seasonal
precipitation are similarly variable.



**Table 7.** Climate means performance: phase II physics tests (i.e. N=12 set 1 changing only RRTMG to New Goddard (NG) and N=12 set 2 changing only land surface model (LSM) from Noah-Unified to Noah-MP (NMP) compared with the phase I physics test RCMs that were shortlisted for further testing (N=12).

| Variable | Timescale | Bias | | | RMSE | | | PSS | | |
|---|---|---|---|---|---|---|---|---|---|---|
| | | Phase I (N=12) ensemble mean | Phase II (NG rad.) ensemble mean | Phase II (NMP LSM) ensemble mean | Phase I (N=12) ensemble mean | Phase II (NG rad.) ensemble mean | Phase II (NMP LSM) ensemble mean | Phase I (N=12) ensemble mean | Phase II (NG rad.) ensemble mean | Phase II (NMP LSM) ensemble mean |
| **Temp. Max. (K)** | Annual | 0.87 | 1.27 | 0.58 | 3.56 | 3.73 | 3.50 | **0.950** | **0.936** | **0.955** |
| | DJF | 0.74 | 1.29 | 0.63 | 4.41 | 4.70 | 4.43 | | | |
| | MAM | 1.40 | 2.06 | 0.83 | 3.68 | 3.92 | 3.55 | - | - | - |
| | JJA | 0.62 | 0.81 | 0.52 | 2.64 | 2.66 | 2.65 | | | |
| | SON | 0.87 | 1.04 | 0.66 | 3.25 | 3.32 | 3.20 | | | |
| **Temp. Min. (K)** | Annual | 1.35 | 0.95 | 1.2 | 3.53 | 3.41 | 3.42 | **0.927** | **0.941** | **0.931** |
| | DJF | 1.50 | 1.08 | 0.87 | 3.86 | 3.82 | 3.66 | | | |
| | MAM | 1.21 | 0.84 | 0.92 | 3.55 | 3.45 | 3.50 | - | - | - |
| | JJA | 0.82 | 0.51 | 0.91 | 3.00 | 2.92 | 3.00 | | | |
| | SON | 1.88 | 1.47 | 1.92 | 3.63 | 3.40 | 3.58 | | | |
| **Prec. (mm)** | Annual | 0.25 | 0.24 | 0.25 | 7.21 | 7.32 | 6.78 | **0.943** | **0.950** | **0.946** |
| | DJF | 0.41 | 0.53 | 0.49 | 8.28 | 8.83 | 8.85 | | | |
| | MAM | 0.32 | 0.32 | 0.25 | 5.91 | 6.47 | 5.53 | - | - | - |
| | JJA | 0.37 | 0.53 | 0.44 | 7.63 | 7.34 | 7.65 | | | |
| | SON | 0.34 | 0.22 | 0.39 | 6.68 | 6.18 | 6.92 | | | |

Phase II RCMs using Noah-MP with RRTMG retained show improved performance in simulating mean maximum and minimum temperature at annual timescales and most seasons relative to corresponding Phase I RCMs using Noah-Unified (Table 7; Figure 4-5). For instance, the mean absolute bias for annual mean maximum temperature is 0.58 K for the Noah-MP ensemble mean versus 1.18 K for the Noah-Unified ensemble. In particular, cold bias magnitudes for maximum temperature are considerably lower over eastern and southern regions for the RCMs using Noah-MP (Figure 4d). RMSE magnitudes for maximum temperature are substantially reduced over the topographically complex regions of the southeast, and southwest and central regions (Figure 5d).

Overall, the magnitude of warm biases for minimum temperature are broadly similar for Phase I and Phase II RCMs (Figure 4b,c). Conversely, while RCMs in both Phases show large RMSEs for minimum temperature over several eastern regions, RMSEs are smaller for the Noah-MP ensemble over some southern areas (Figure 5b,c).





In contrast to the above results for the simulation of maximum temperature, overall, Phase II
RCMs using Noah-MP show smaller performance improvements for the simulation of precipitation
relative to the Phase I RCMs (Table 7). However, precipitation bias magnitudes are smaller for the
Noah-MP ensemble over specific regions, e.g. north-eastern coastal regions and the elevated terrain of
the south-east (Figure 4c,f).

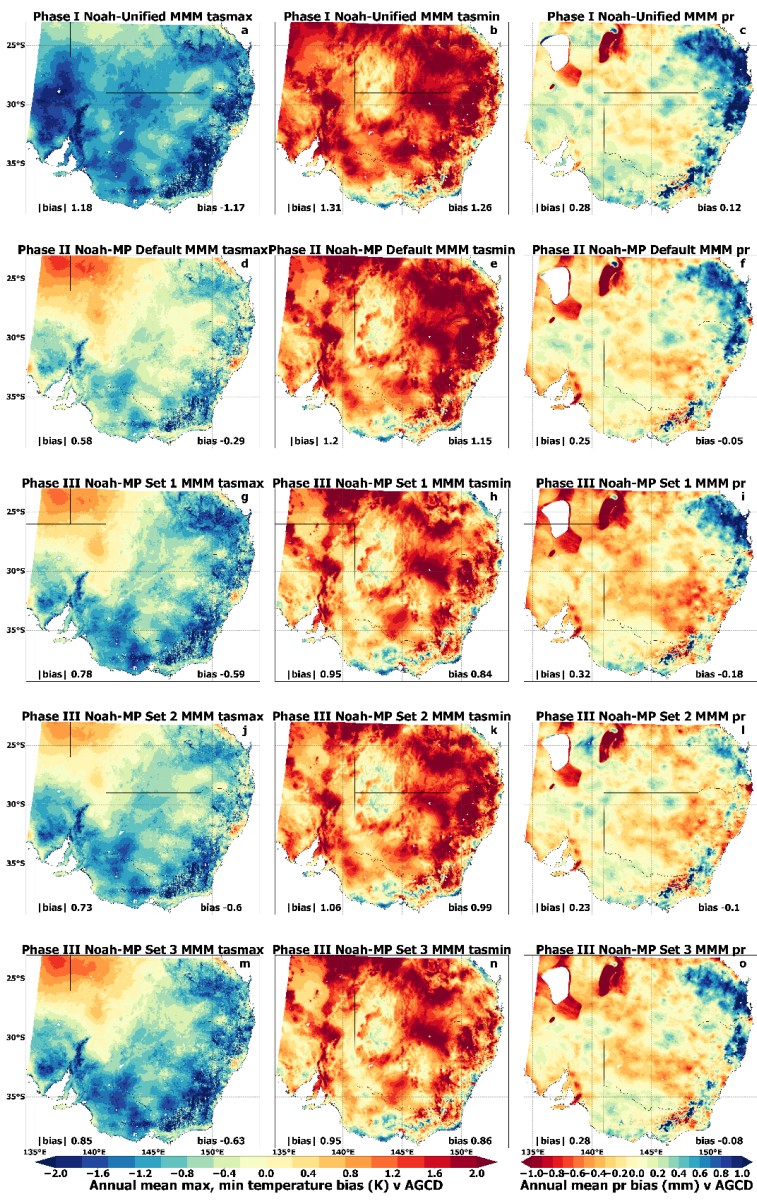


**Figure 4.** Phase I (N=36), Phase II (N=60) and Phase III (N=78) ensemble mean biases for annual mean maxi-
mum temperature, minimum temperature and precipitation with respect to Australian Gridded Climate Data





(AGCD) observations for NARCliM2.0 Phase I physics test RCMs using Noah-Unified as the land surface
model (LSM) (a-c); Phase II physics test RCMs using Noah-MP as the LSM and its default settings (d-f); Phase
III 'set 1' physics test RCMs using Noah-MP with dynamic vegetation cover activated (g-i); Phase III 'set 2'
physics test RCMs using Noah-MP with TOPMODEL surface runoff and simple groundwater activated (j-l);
and Phase III 'set 3' physics test RCMs using Noah-MP with both dynamic vegetation cover and TOPMODEL
runoff activated (m-o).

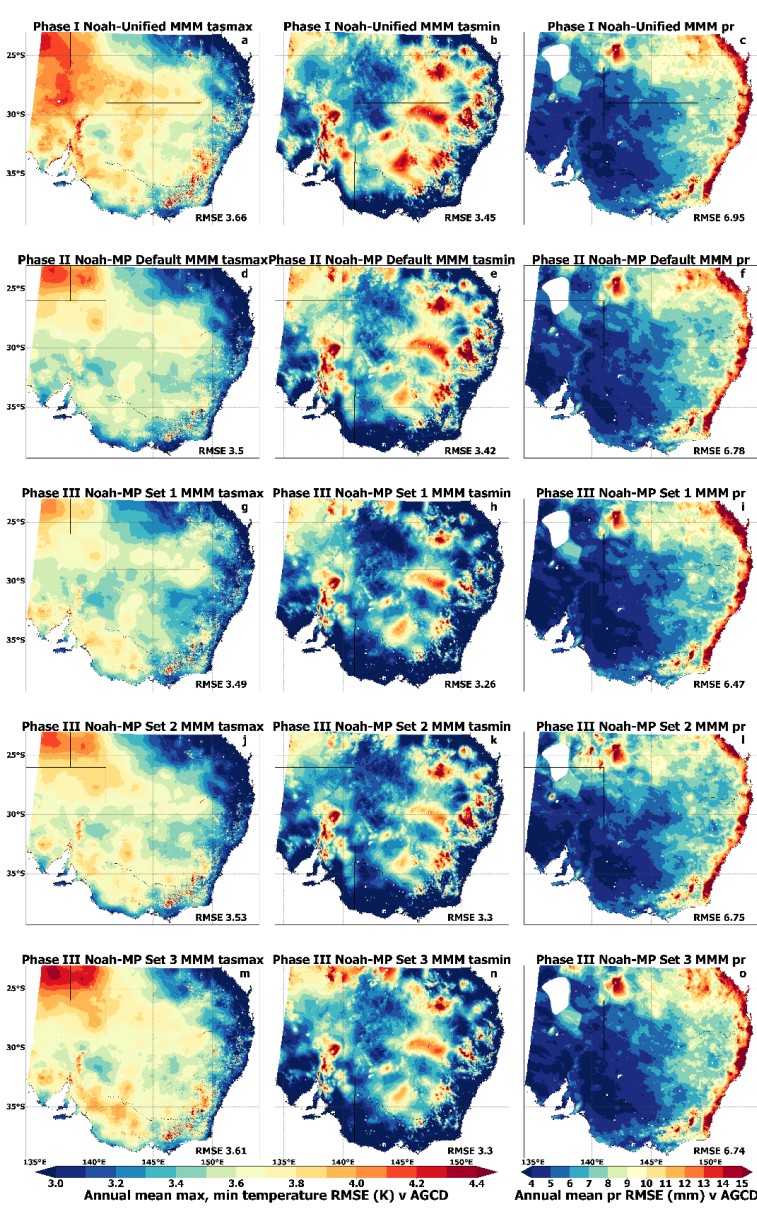


**Figure 5.** As per Figure 4 but showing RMSEs.





### 5.2.2. Climate Extremes

Climate extreme analysis assesses RCM representations of the hottest and the wettest day versus AGCD. For both extremes and for RCM biases and RMSEs, Phase II RCMs using NG radiation showed inferior performance relative to phase I RCMs using RRTMG (Table 8). Conversely, Phase II RCMs using Noah-MP show substantial reductions in bias for both the hottest and wettest days (Table 8). Phase II Noah-MP RCMs show a small increase in RMSE for the hottest day (Phase I bias=3.59 K; Phase II bias=3.74 K); however, RMSEs are smaller for the wettest day (i.e. Phase I RMSE=19.20 mm; Phase II RMSE=18.47 mm) (Table 8).

**Table 8** Climate extremes performance: comparing phase I RCMs (N=12) with phase II RCMs (i.e. 12 RCMs changing radiation from RRTMG to New Goddard (NG) and 12 RCMs changing land sur-face model (LSM) from Noah-Unified to Noah-MP; NMP).

| Variable | Bias | | | RMSE | | |
|---|---|---|---|---|---|---|
| | Phase I (N=12) ensemble mean | Phase II (NG rad.) ensemble mean | Phase II (NMP LSM) ensemble mean | Phase I (N=12) ensemble mean | Phase II (NG rad.) ensemble mean | Phase II (NMP LSM) ensemble mean |
| **Temp. max: hottest (K)** | 1.11 | 1.93 | 0.81 | 3.59 | 3.97 | 3.74 |
| **Prec.: wettest (mm)** | 3.08 | 3.21 | 2.60 | 19.20 | 20.52 | 18.47 |

## 5.3 Phase III RCM performance summary and shortlisting N=7 RCMs for ERA5-NARCliM2.0 evaluation simulations

Overall, RCM biases for mean maximum temperature do not show marked improvements once the dynamic vegetation cover and surface runoff options are activated for Noah-MP (Figure 4 g,j,m) rela-tive to RCMs using Noah-MP with default settings (Figure 4d). However, specifically for the RCM ensemble with dynamic vegetation cover activated for Noah-MP, RMSE magnitudes for maximum temperature are lower over some eastern coastal regions (Figure 5g).

The simulation of mean minimum temperature shows clear performance improvements for Phase III RCMs using options activated for Noah-MP, relative to RCMs using Noah-MP defaults. Overall, both biases and RMSEs for minimum temperature are reduced in magnitude for RCMs using the either or both of dynamic vegetation cover and runoff/groundwater options activated for Noah-





MP, relative to the default parameters (Figure 4-5). These performance improvements are largest over
eastern and southern regions.

There are no substantial overall performance improvements in the simulation of precipitation

for Phase III RCMs relative to Phase II RCMs (Figures 4-5 f,i,l,o). However, using Noah-MP with
specific LSM options remains favourable to using RCMs with Noah-Unified, albeit the performance
gains are generally small, except for some coastal regions and especially the north-east.

All 78 RCMs in the complete RCM physics test ensemble are ranked for performance as de-

scribed in sect. 4.2. Once the poor-performing RCMs are excluded, there are 20 RCMs remaining
(Table 9; Figures 6-8). In Table 9, we see that 16 Noah-MP-based RCMs from Phase II and Phase III
comprise this set of 20 RCMs, with 3 of the 20 RCMs using Noah-Unified, and 1 using CLM4.0. For
maximum temperature, some shortlisted RCMs show large RMSEs over north-western and inland
areas (e.g. Figure 6 d-f) that are of similar magnitude to those of the ensemble means of Phase I-III
RCMs (Figure 5). Conversely, several shortlisted RCMs show very low RMSEs for maximum tem-
perature across eastern and southern regions, especially along the eastern coast (Figure 6, e.g. RCMs
in panels d,l,n,o,q). For minimum temperature, a subset of the twenty shortlisted RCMs show substan-
tially reduced RMSEs over many regions relative to the Phase I-III ensemble means (Figure 7, e.g.
RCMs in panels: b,h,i). Additionally, several shortlisted RCMs show reduced RMSEs for precipita-
tion over the eastern coast and north-east (Figure 8, e.g. RCMs in panels: c, l, m, n, o) relative to the
Phase I-III RCM ensemble means in Figure 5c,f,i,l,o.

These 20 RCMs are assessed for statistical independence and 7 RCMs from this RCM set are

shortlisted for the ERA5-forced RCM simulations considering both their performance and independ-
ence scores (Table 9). These 7 shortlisted RCMs are listed in **bold** in Table 9 and are identified as R1-
R7 in the ERA5-forced evaluation simulations (Table 9; final column). RCMs are shortlisted from the
set of 20 if they rank highly for both performance and independence. For instance, RCM
050801040404_set_3 (top row, Table 9) is top-ranked for performance, however, its independence
scores/ranks are low, hence it is not shortlisted. It is important to note that, while a general perfor-
mance gain is observed in the physics testing when using Noah-MP, there are some specific RCM
configurations using Noah-Unified that perform well in simulating the Australian climate. For in-
stance, the RCM 010602050502 (row 7; Table 9; 'R1') uses Noah-Unified and performs well overall
(its overall performance rank=7), and especially for the simulation of maximum temperature (Figure
6a). It is also the only RCM in this set of 20 RCMs to use YSU for PBL. Importantly, this RCM is
highly ranked for statistical independence, hence, this RCM is shortlisted for the N=7 set.





**Table 9**. The 20 NARCliM2.0 physics test RCMs shortlisted from the full ensemble of 78 RCMs based on their
performance in simulating the Australian climate and independence (Ind.) scores. N=7 'R1-R7' RCMs shortlist-
ed for ERA5-forced CORDEX evaluation simulations shown in **bold**. NU=Noah Unified; NMP=Noah-MP;
DV=dynamic vegetation cover; TOP=topmodel runoff.

| # | RCM Physics Combination | PBL | MP | Cumulus | SW/LW | LSM | Test Phase | Overall Performance Rank | Bishop Abramowitz Ind. Rank | Herger Ind. Set 1 | Herger Ind. Set 2 | ERA5-forced RCM Identifier |
|---|---|---|---|---|---|---|---|---|---|---|---|---|
| 1 | 050801040404_set_3 | MYNN2 | Thom | KF | RRTMG | NMP DV+TOP | III | 1 | 19 | 20 | 20 | |
| **2** | **070806040404_set_1** | **ACM2** | **Thom** | **Td** | **RRTMG** | **NMP DV** | **III** | **2** | **8** | **5** | **6** | **R6** |
| 3 | 50801040404 | MYNN2 | Thom | KF | RRTMG | NMP | II | 3 | 16 | 12 | 13 | |
| **4** | **070802040404_set_1** | **ACM2** | **Thom** | **BMJ** | **RRTMG** | **NMP DV** | **III** | **4** | **4** | **3** | **3** | **R5** |
| 5 | 070802040404_set_2 | ACM2 | Thom | BMJ | RRTMG | NMP TOP | III | 5 | 15 | 13 | 12 | |
| **6** | **050601040404_set_1** | **MYNN2** | **WSM6** | **KF** | **RRTMG** | **NMP DV** | **III** | **6** | **7** | **10** | **10** | **R2** |
| **7** | **10602050502** | **YSU** | **WSM6** | **BMJ** | **NG** | **NU** | **II** | **7** | **1** | **3** | **3** | **R1** |
| **8** | **070806040404_set_2** | **ACM2** | **Thom** | **Td** | **RRTMG** | **NMP TOP** | **III** | **8** | **9** | **9** | **5** | **R7** |
| 9 | 70806040404 | ACM2 | Thom | Td | RRTMG | NMP | II | 9 | 11 | 14 | 14 | |
| # | 50802040404 | MYNN2 | Thom | BMJ | RRTMG | NMP | II | 10 | 20 | 19 | 19 | |
| **#** | **050802040404_set_1** | **MYNN2** | **Thom** | **BMJ** | **RRTMG** | **NMP DV** | **III** | **11** | **5** | **2** | **2** | **R3** |
| # | 070806040404_set_3 | ACM2 | Thom | Td | RRTMG | NMP DV+TOP | III | 14 | 12 | 10 | 10 | |
| # | 70802040404 | ACM2 | Thom | BMJ | RRTMG | NMP | II | 17 | 13 | 15 | 15 | |
| # | 070601040404_set_3 | ACM2 | WSM6 | KF | RRTMG | NMP DV+TOP | III | 22 | 14 | 16 | 16 | |
| **#** | **050802040404_set_2** | **MYNN2** | **Thom** | **BMJ** | **RRTMG** | **NMP TOP** | **III** | **23** | **2** | **4** | **4** | **R4** |
| # | 70802050502 | ACM2 | Thom | BMJ | NG | NU | II | 24 | 18 | 18 | 18 | |
| # | 50801040405 | MYNN2 | Thom | KF | RRTMG | CLM4 | I | 28 | 17 | 17 | 17 | |
| # | 070601040404_set_1 | ACM2 | WSM6 | KF | RRTMG | NMP DV | III | 29 | 6 | 7 | 8 | |
| # | 70801040404 | ACM2 | Thom | KF | RRTMG | NMP | II | 30 | 3 | 1 | 1 | |
| # | 50801040402 | MYNN2 | Thom | KF | RRTMG | NU | I | 31 | 10 | 6 | 7 | |



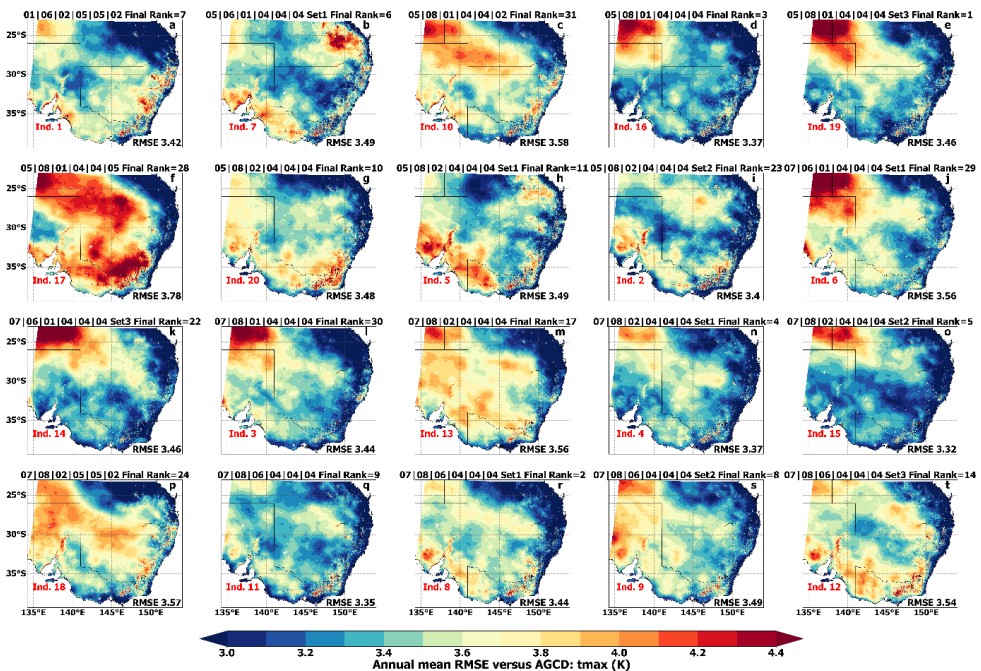

**Figure 6.** RMSEs for modelled mean maximum temperature (tmax) versus observations for the twenty
NARCliM2.0 physics test RCMs shortlisted from the full ensemble of seventy-eight RCMs based on their
performance in simulating the recent south-east Australian climate. Overall (final) performance ranks and
Bishop and Abramowitz (2013) method independence (Ind.) scores are shown.

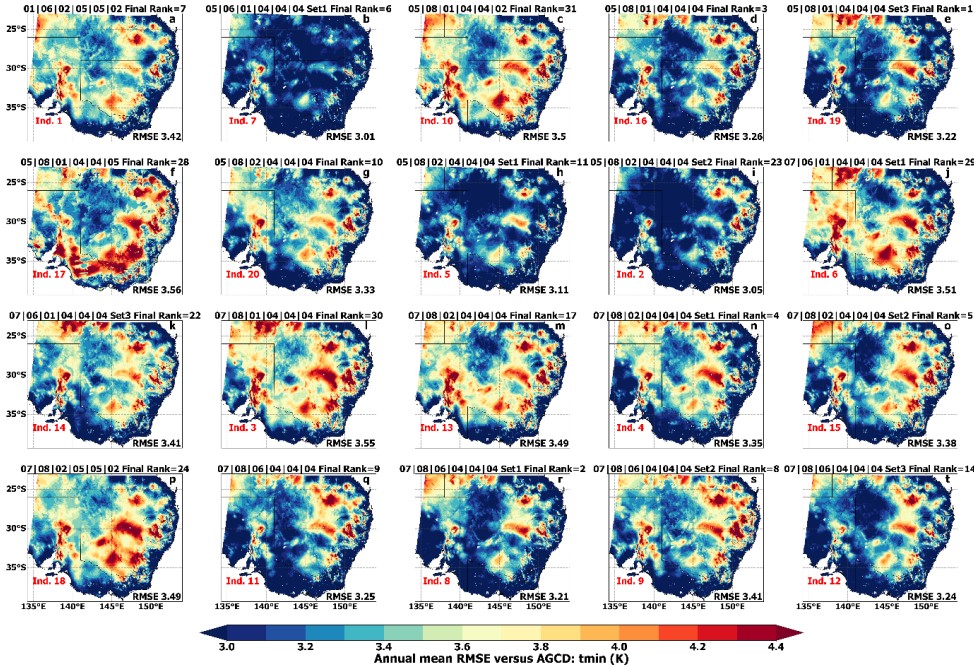

**Figure 7.** As per Figure 6 but for mean minimum temperature (tmin).

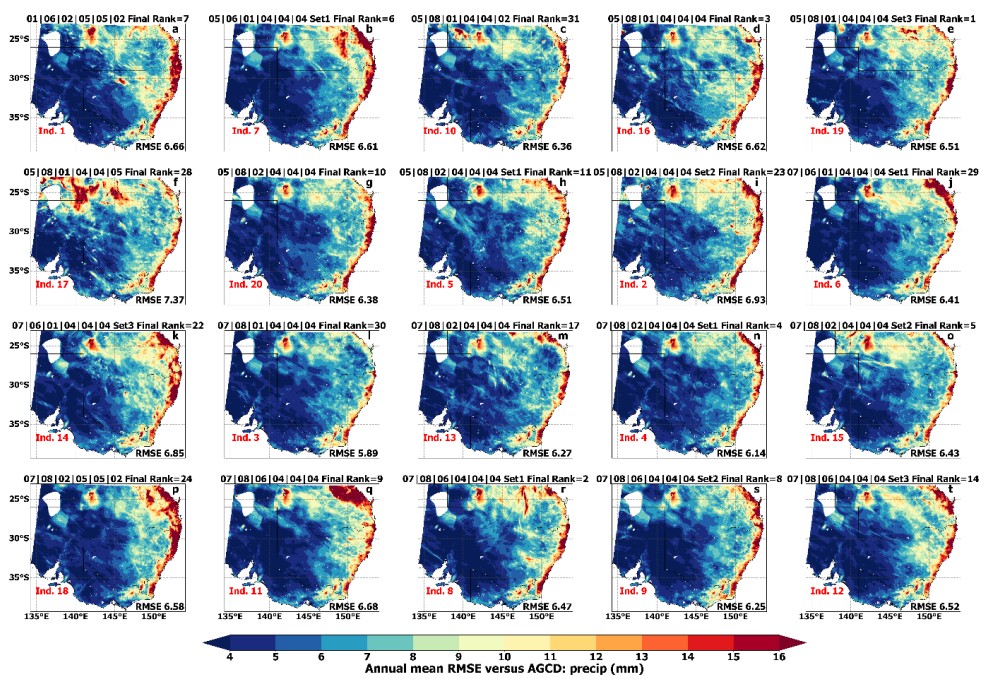

**Figure 8.** As per Figure 6 but for mean precipitation (precip.).

## 6. CORDEX-CMIP6 NARCliM2.0 historical evaluation

### 6.1 Maximum temperature

NARCliM2.0 RCMs simulate maximum temperature more accurately than NARCliM1.x, with widespread, statistically significant reductions in cold biases (Figure 9). These reductions in bias apply for all timescales but are largest for the annual mean, i.e. the area-averaged mean absolute bias is 0.75°C for the NARCliM2.0 ensemble, 1.73°C for NARCliM1.5, and 1.89°C for NARCliM1.0 (Figure 9d,g,j). Notably, annual mean maximum temperature bias magnitudes are very small, i.e. around <0.5°C, over south-west WA, southern coastal regions, and several eastern regions. This may be important from a climate change adaptation and mitigation perspective as these regions are heavily populated and economically significant. NARCliM2.0 retains warm biases of similar magnitude to NARCliM1.5 along the north-west coast of Australia (Figure 9d,g). Moreover, these warm biases cover additional areas for NARCliM2.0, especially during DJF (Figure 9e,h). A wide range of bias signs are evident for the individual NARCliM2.0 ensemble members (Figures S4-S6). The R5 RCM is generally warmer than R3, e.g. (Figure S4c,d). Considering the forcing GCM data, overall, ensemble means for the CMIP6 and CMIP5 GCMs generally show similar patterns and magnitudes of cold bias for maximum temperature (Supporting Information S7).

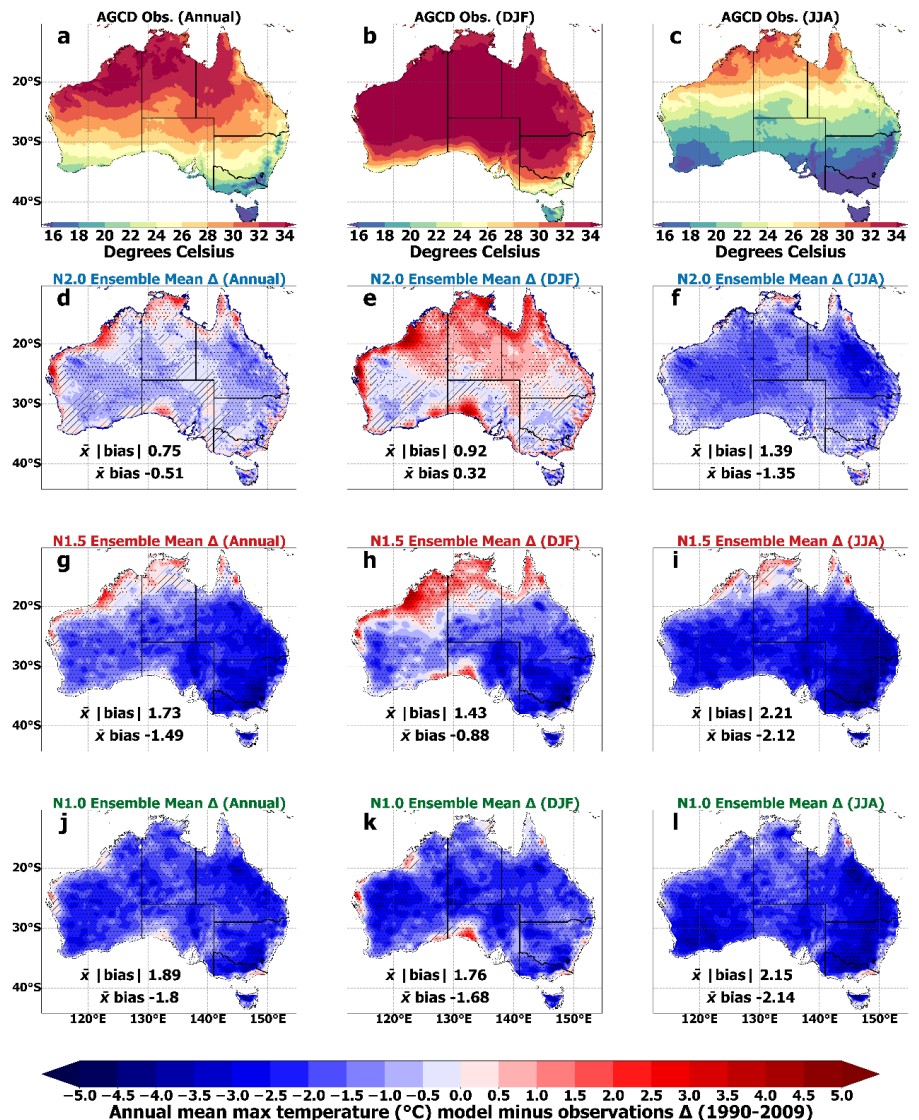

**Figure 9.** Annual, DJF and JJA mean near-surface atmospheric maximum temperature biases for NARCliM2.0,
1.5 and 1.0 historical ensemble means with respect to Australian Gridded Climate Data (AGCD) observations
for 1990-2009. Stippled areas indicate locations where an RCM shows statistically significant bias (*P*<0.05).
Significance stippling for the ensemble mean bias follows Tebaldi et al. (2011) and is applied separately to each
RCM ensemble. Statistically insignificant areas are shown in colour, denoting that less than half of the models
are significantly biased. In significant agreeing areas (stippled), at least half of RCMs are significantly biased,
and at least 70% of significant RCMs in each ensemble agree on the direction of the bias. Significant disagree-
ing areas are shown in hatching, which are where at least half of the models are significantly biased and less
than 70% of significant models in each ensemble agree on the bias direction - see main text for additional detail
on the stippling regime.





## 6.2 Minimum temperature

The simulation of mean minimum temperature by NARCliM2.0 is generally warm biased at all time-
scales (Figure 10). Its bias magnitudes over many regions are larger versus NARCliM1.5, e.g. annual
mean area-averaged absolute biases are 0.98°C and 0.79°C for NARCliM2.0 and NARCliM1.5, re-
spectively (Figure 10 d,g). However, there are exceptions to this result over specific regions, for ex-
ample, parts of south-west western Australia show annual mean bias magnitudes of <1°C for NAR-
CliM2.0, but these areas show biases below -2°C for NARCliM1.x (Figure 10d,g,j). Most individual
RCMs comprising the NARCliM2.0 ensemble show stronger warm biases than their NARCliM1.5
peers at both annual and seasonal timescales (Figures S8-S10). The ACCESS-ESM-1-5-forced NAR-
CliM2.0 RCMs are considerably more warm-biased than the other NARCliM2.0 RCMs, with average
absolute biases of 1.74°C and 1.9°C; Fig. S8c-d).
Many of the CMIP6 GCMs used to force the NARCliM2.0 RCMs are warmer than the CMIP5
GCMs used to force NARCliM1.5, such that the ensemble mean bias of the former is 1.9°C versus
1.11°C (Figure S11). In particular, ACCESS-ESM-1-5 and MPI-ESM1-2-HR are substantially more
warm-biased relative to all other selected GCMs, with mean absolute biases of 2.2°C and 3.47°C, re-
spectively (Figure S11). This suggests that NARCliM2.0's warm biases for mean minimum tempera-
ture are at least partially inherited from the driving data. However, whilst the ACCESS-ESM-1-5-
forced NARCliM2.0 RCMs are much warmer than their counterparts (i.e. 1.74°C and 1.9°C), this
does not apply to the MPI-ESM1-2-HR-forced RCMs, which have biases of only 1.01°C and 1.09°C.
Hence, factors additional to the driving data, such as changes in RCM parameterisations between
NARCliM generations and other model design changes likely contribute to the warmer biases ob-
served for NARCliM2.0.

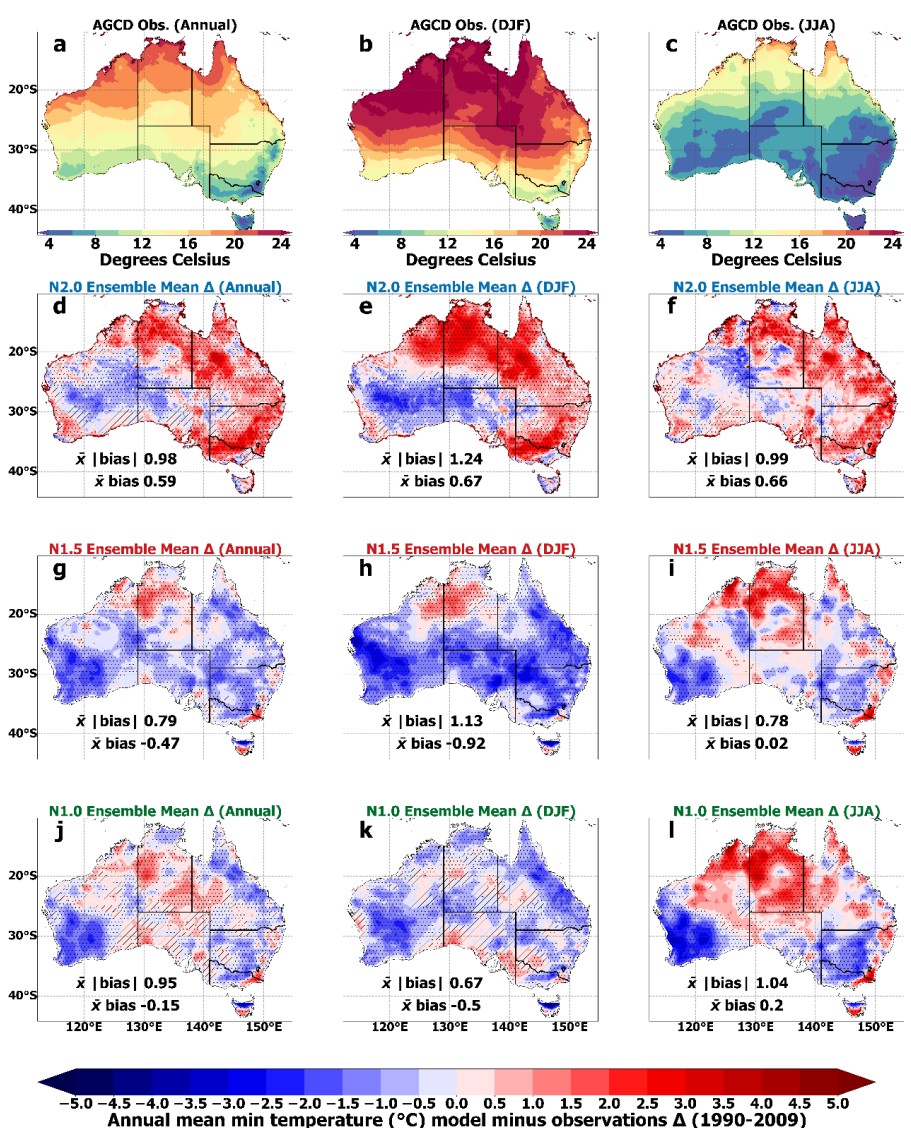

**Figure 10.** As per Figure 9 but for mean minimum temperature.

## 6.3 Precipitation

The NARCliM2.0 ensemble shows small dry biases for mean precipitation over most regions, except
for some areas mainly in the east of the country which show slight wet biases (Figure 11d-f). This
contrasts with stronger, statistically significant wet biases of NARCliM1.5 (Figure 11g-i) and the even
stronger wet biases of NARCliM1.0 (Figure 11j-l). Area-averaged bias magnitudes are considerably
smaller for NARCliM2.0 relative to NARCliM1.x, especially for the annual mean, i.e. 8.03 mm ver-
sus 16.69 mm and 33.25 mm, respectively. Annual mean precipitation biases are particularly small



over eastern regions, often being <5 mm. NARCliM2.0 retains the strong summertime dry biases for precipitation over northern Australia that are evident for NARCliM1.5 (Figure 11e,h), noting that this region also shows strong warm biases for maximum temperature (Figure 9).

The individual RCMs comprising NARCliM2.0 show a range of results for annual and seasonal mean precipitation biases (Fig S12-S14). Notably, three of the ten NARCliM2.0 RCMs have substantially larger bias magnitudes than their peers at annual and summer timescales, i.e. both MPI-ESM1-2-HR-R3 and R5 (absolute biases are 15.53 mm and 22.45 mm for annual mean precipitation, Figure S12g-h) and EC-Earth3-Veg-R5 (Figure S12f; 18.59 mm). Despite EC-Earth3-Veg-R5 being strongly dry-biased, EC-Earth3-Veg-R3 simulates precipitation more accurately i.e. its mean absolute bias=9.53 mm (Figure S12e). Analogously to NARCliM2.0's performances for temperature, R5 is drier than R3. Comparing the ensemble means of the driving GCMs, the CMPI6 GCMs are marginally more accurate in simulating annual mean precipitation than the CMIP5 GCMs (Figure S15). Whilst the CMIP6 ensemble produces small biases over inland regions, its biases are larger along the east coast.

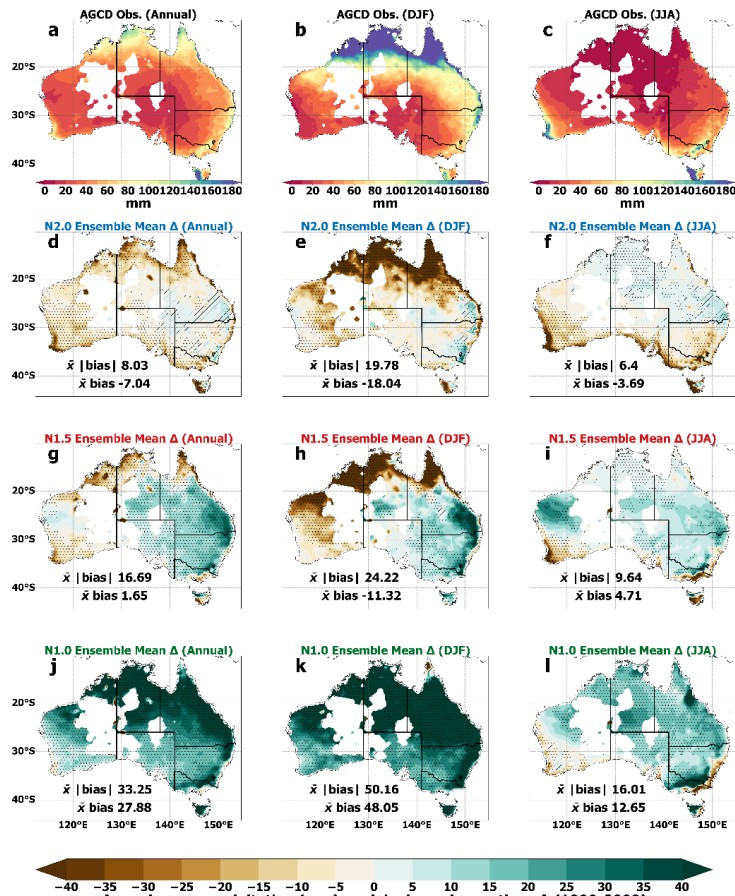

**Figure 11.** As per Figure 9 but for mean precipitation (precip.).





# 7. CORDEX-CMIP6 NARCliM2.0 climate change projections

Dependent on location, the largest maximum temperature projected increases for NARCliM2.0 under SSP3-7.0 are over ~3°C, and over ~1.5°C under SSP1-2.6 (Figure 12a,d). SSP3-7.0-NARCliM2.0 shows faster warming over inland than coastal regions, with greater warming across a horizontal band of the continent during annual and summer timescales (Figure 12a-b). This contrasts with NAR-CliM1.5 which shows a north-south warming gradient at annual and seasonal timescales, with its fastest warming rate over northern regions, and NARCliM1.0 which projects fastest warming over the west (Figure 12). For NARCliM2.0, the tropical north warms faster during the winter dry season than during the summer wet season under SSP3-7.0, but this is not the case for SSP1-2.6 (Figure 12b-c; e-f). NARCliM2.0 simulations under SSP3-7.0 show less warming than NARCliM1.5-RCP8.5, but warmer futures than for NARCliM1.0-SRES A2, with differences in the underlying driving GCMs and GHG scenarios likely contributing to these variations in warming. As per NARCliM1.x, all NARCliM2.0 maximum temperature projections are significant-agreeing with all RCMs projecting statistically significant temperature increases.

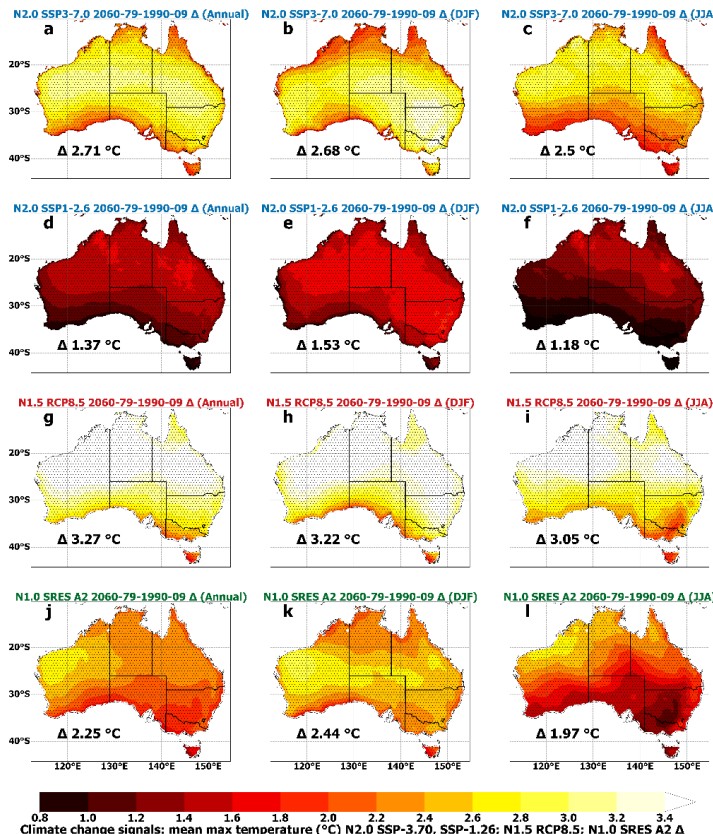

**Figure 12.** Ensemble mean climate change projections (far future minus present-day) for annual, DJF and JJA mean maximum temperatures with significance stippling as per Figure 9.





Projected increases in annual mean minimum temperature for NARCliM2.0 exceed 3°C over
some regions for SSP3-7.0, and 1.6°C for SSP1-2.6 (Figure 13). Under both GHG scenarios, at annual
and winter timescales warming is fastest over north-east Australia. Conversely, NARCliM1.x mini-
mum temperature future increases are generally largest over northwest or northern Australia, though
the summertime projection for NARCliM1.0 is an exception (Figure 13k). As for maximum tempera-
ture projections, all RCMs for all NARCliM generations project statistically significant increases.

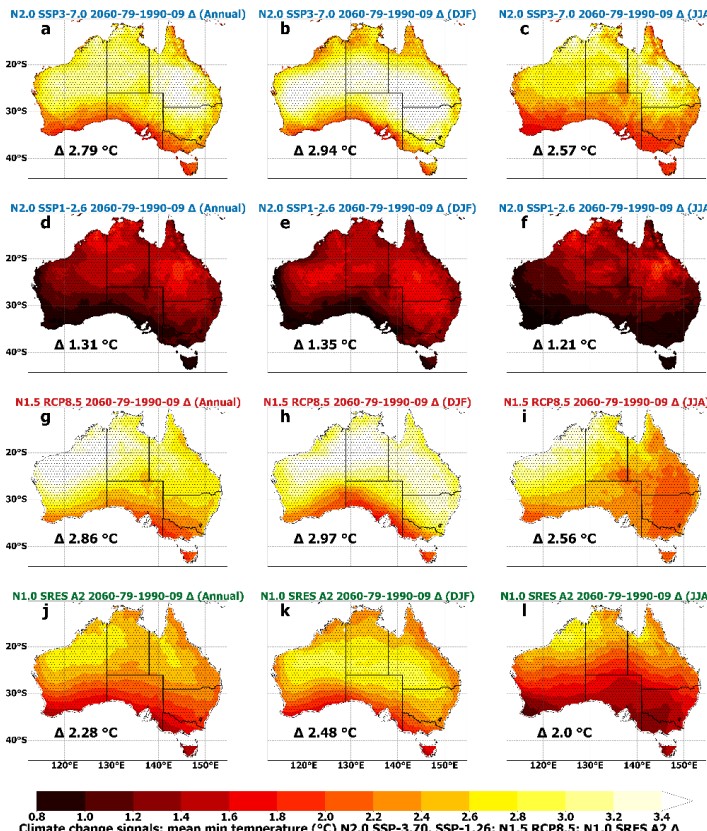


**Figure 13.** Ensemble mean climate change projections (far future minus present-day) for annual, DJF and JJA

mean minimum temperatures with significance stippling as per Figure 9.
NARCliM2.0 SSP3-7.0 projects a dry future over most of Australia, except for wetter futures
over northern and western regions, which are largest in magnitude in summer (Figure 14a-b). In con-
trast, overall, NARCliM2.0 SSP1-2.6 projects dry changes across most of Australia, with the strongest
drying over northern Australia during summer (Figure 14e). Similarities between NARCliM2.0 pro-
jections for the low and high GHG SSPs include faster drying over the eastern coastline at all time-
scales, especially during summer. The wetter futures projected by RCMs downscaling SSP3-7.0-
GCMs relative to SSP1-2.6 may be partially inherited from the driving CMIP6 GCMs, because over-
all, SSP3-7.0 GCMs show wetter futures than corresponding SSP1-2.6 GCMs (Fig. S16).



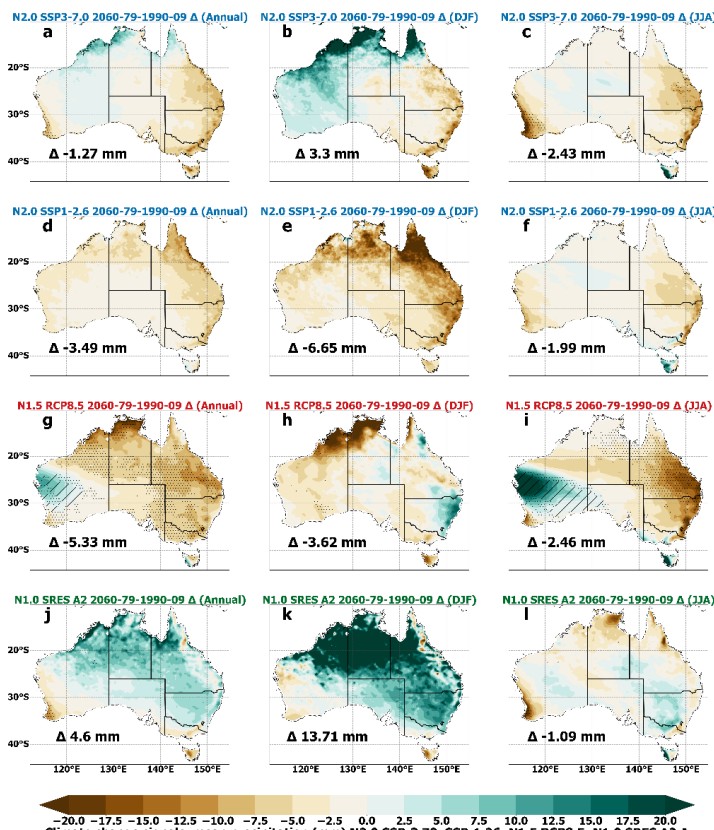

**Figure 14.** Ensemble mean climate change projections (far future minus present-day) for annual, DJF and JJA mean precipitation with significance stippling as per Figure 9.

Considering mean precipitation projections for individual NARCliM2.0 RCMs, in some cases, R3 and R5 RCMs produce similar results when downscaling the same GCM. For instance, ACCESS-ESM-1-5 forced R3 and R5 both show statistically significant projected decreases in annual mean precipitation across Australia (Figure 15b-c). In contrast, while UK-ESM1-0-LL R3-R5 both show projected decreases in annual mean precipitation over eastern Australia, R3 shows precipitation increases that are substantially more widespread over western and northern regions relative to R5 (Figure 15j-k). Overall, the NARCliM2.0 ensemble members show a variety of climate change signals for precipitation (Figure 15) and temperature (not shown), reflecting the range within the larger CMIP6 ensemble (Di Virgilio et al. 2022).

There are some key differences between the mean precipitation projections of NARCliM2.0 relative to those of previous NARCliM generations. For instance, NARCliM1.5 shows stronger reductions in future precipitation over northern and eastern regions at annual and winter timescales (Figure 14), and these changes are statistically significant over many regions, whereas there are only small regions of significant changes for NARCliM2.0. Additionally, NARCliM2.0 projects marked precipi-



tation decreases along the south-east coast during summer, while NARCliM1.5 shows the opposite

result (Figure 14). NARCliM1.0 generally projects wet futures across larger portions of Australia,

especially at annual and summer timescales.

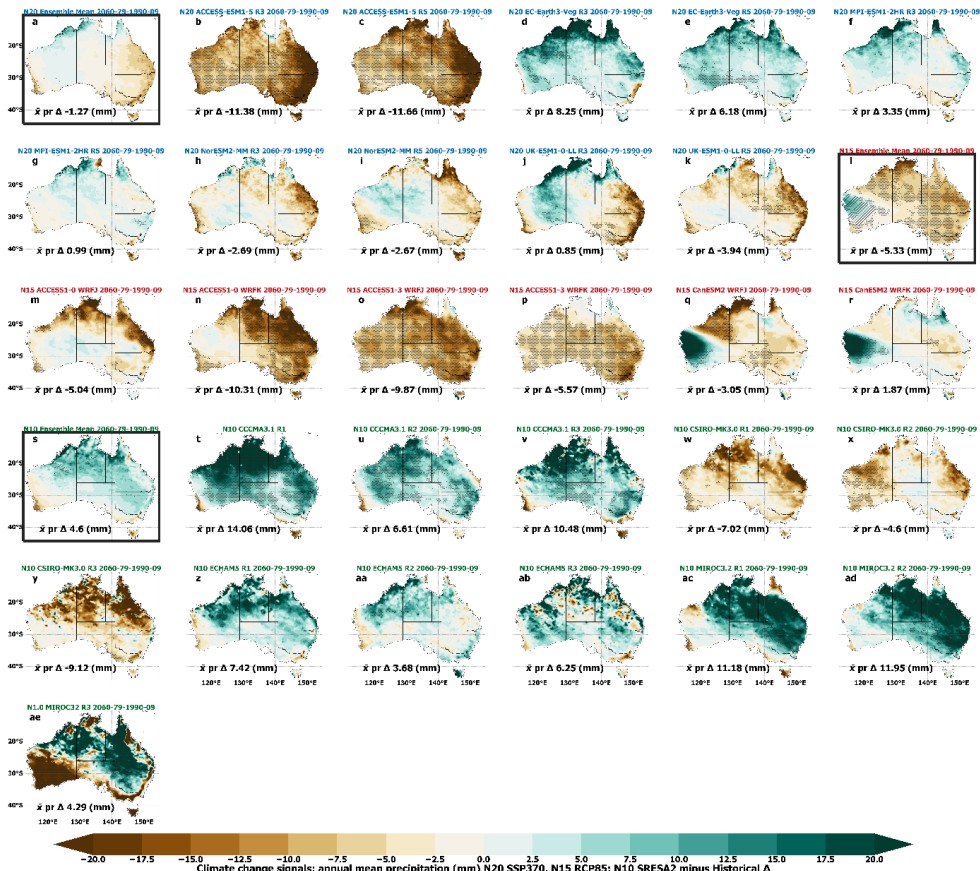

**Figure 15.** Climate change projections (1990-2009 versus 2060-2079) for annual mean precipitation for NAR-
CliM ensemble mean climate change signals (a,l,s) and for individual ensemble members. Significance stippling
as per Figure 9.

## 8. Discussion and Summary

NARCliM regional climate models produce robust climate projections at spatial scales suitable for

local-scale climate change analysis and impact decision-making. The third and latest generation of

these regional climate models, NARCliM2.0, encompasses several model design advancements over

its predecessors.





## 8.1 NARCliM2.0 RCM physics testing

A key aim of this paper is to describe how NARCliM2.0 differs from its predecessors and explain the rationale for these design decisions. In addition to RCM design choices including increased resolution, and incorporation of convection-permitting modelling and urban physics, a major change for NARCliM2.0 relative to its predecessors is to use new WRF RCM configurations which are selected via a large suite of physics tests. RCM performance evaluations for the NARCliM2.0 RCM physics testing focused on the 4 km resolution convection-permitting domain which does not use a cumulus physics parameterisation. Notably, the 7 'candidate' shortlisted RCMs from the N=78 physics test ensemble used three different cumulus parameterisations for their outer domains, with 4 RCMs using BMJ, 2 RCMs using Tiedtke, and 1 using Kain-Fritsch. This indicates that differences in the outer domain boundary conditions have key influences on the RCM performances in the convection-permitting domain.

The use of the Noah-MP LSM in the NARCliM2.0 RCM physics tests conferred overall RCM skill improvements relative to the test Phase I RCMs using the Noah-Unified LSM, especially in terms of the simulation of temperature. Although using Noah-MP also improved the simulation of precipitation in some respects, these improvements were smaller relative to the gains for temperature, and improvements were mainly located over coastal regions. The developers of Noah-MP suggest that some limitations in the Noah-Unified LSM have been modified to better represent several parameters. These include surface layer radiation balances, snow depth, soil moisture and heat fluxes, leaf area-rainfall interaction, vegetation and canopy temperature distinction, drainage of soil, and runoff.

In the NARCliM2.0 physics testing, improvements in RCM skill were evident for Noah-MP with default settings. Activating specific parameterisations for this LSM (i.e. dynamic vegetation cover and surface runoff-simple groundwater) delivered comparatively smaller gains in RCM performances. Some previous studies have found no overall benefit of using Noah-MP with default settings. For instance, Imran et al. (2018) conducted an evaluation of WRF coupled with a variety of LSMs including Noah-MP using its default settings. Their focus was on simulating short-duration (~3-day) heatwaves in Melbourne, Australia. They observed larger temperature biases using Noah-MP relative to RCMs using Noah-Unified and CLM4.0. However, their focus on specific heatwave events of short duration over one urban area was not intended as a comprehensive evaluation of Noah-MP's performance using default settings over longer timescales. It is also important to consider that several physics schemes used by these authors differed to those used in the NARCliM2.0 physics testing, i.e. they used: PBL=MYJ; microphysics=Thompson; cumulus=Grell3D; radiation=RRTMG/RRTMG. The only similarities between these settings and those of the NARCliM2.0 physics testing are the use of Thompson microphysics and RRTMG. WRF and Noah-MP versions also differed, i.e. Imran et al. used WRF3.6.1 and a Noah-MP version prior to 3.7, whereas NARCliM2.0 uses WRF4.1.2 and Noah-MP version 4.1.





In an assessment of the performances of several WRF-LSMs for Sub-Saharan Africa, Glotfelty et
al. (2021) noted deficiencies in the simulation of land use and land cover change (LULCC) parame-
ters such as surface albedo by Noah-MP. Despite these deficiencies, the spatial patterns and magni-
tudes of temperature and precipitation were well-represented by Noah-MP. However, the land surface
parameter errors impacted the magnitude and sign of LULCC-induced changes in temperature and
precipitation. These deficiencies were linked to substantial underestimations of surface albedo in arid
areas due to inaccurate soil albedo treatments by Noah-MP. Moreover, errors in Noah-MP's LAI pro-
files may occur because it was developed principally for application in Northern Hemisphere mid-
latitudes. It is possible that modifying/tuning Noah-MP to specific aspects of the Australian context
would yield performance benefits for follow-up dynamical downscaling. Overall, these authors con-
cluded that "Noah-MP is least flawed of the [WRF] default LSMs". Additionally, there are also sever-
al studies that have reported benefits of using Noah-MP with default parameters relative to other
LSMs e.g. Chen et al. (2014b), Chen et al. (2014a) and Salamanca et al. (2018).
The NARCliM2.0 physics testing found that the optimal LSM configuration for simulation of
minimum temperature used Noah-MP with dynamic vegetation cover activated, even though the per-
formance gain relative to Noah-MP with default settings was small. Constantinidou et al. (2020) ran
WRF coupled with four LSMs (Noah-Unified, Noah-MP, CLM and, Rapid Update Cycle) over Mid-
dle East North Africa CORDEX domain. Their study compared the performance of Noah-MP with
dynamic vegetation cover turned on and off. They showed that air and land temperatures were best
simulated using Noah-MP with dynamic vegetation cover activated.
Overall, Noah-MP performed well in the NARCliM2.0 physics tests, conferring some clear ad-
vantages over RCMs using Noah-Unified. However, given the nature of its development and perfor-
mance characteristics, it may be more suited to application over the temperate regions of Australia
rather than the semi-arid interior.
In terms of PBL parameterisations, by the completion of Phase I physics testing, only 3 of 12
RCMs shortlisted for further testing use the YSU scheme. By the completion of Phase II testing, all
remaining RCMs using YSU are discarded, with only RCMs using PBL schemes other than YSU re-
maining (i.e. ACM2 and MYNN2). YSU PBL is a first-order closure scheme that expresses turbulent
mixing via mean variables rather than prognostic variables (Hong et al., 2006). It is classed as a 'non-
local' scheme because it estimates turbulent mixing by small-scale eddies as well as representing
transport caused by convective large eddies. Two previous studies evaluating convection permitting
WRF simulations using different parameterisations that included YSU for the PBL scheme found that,
relative to other PBL schemes, YSU produced the highest bias for simulated precipitation (Huang et
al., 2023; Nuryanto et al., 2019). However, these studies focused on different regions globally, and
used various experimental setups that are not directly comparable to those used here. Hence, a sepa-
rate study investigating sensitivities of the NARCliM2.0 RCMs to the different PBL schemes is cur-
rently underway.





## 8.2 CMIP6-NARCliM2.0: historical evaluation and climate change projections

We characterised the improvements conferred by NARCliM2.0 over its predecessors in simulating the present-day Australian climate. NARCliM2.0 simulates mean maximum temperature and precipitation more accurately than NARCliM1.x. Specifically, NARCliM1.x has strong maximum temperature cold biases which are in keeping with other downscaling projects of the CMIP3-CMIP5 eras, e.g. (Andrys et al., 2016; Evans et al., 2020b), but these are substantially reduced in NARCliM2.0. A contributing cause of CMIP5-forced RCM cold biases of maximum temperature is their overestimation of precipitation (Evans et al., 2020). This relationship was also noted in ERA-Interim forced RCMs of this modelling era (Di Virgilio et al. 2019). In NARCliM2.0, the widespread wet biases that characterise the NARCliM1.x RCMs are greatly reduced in magnitude. NARCliM2.0 produces smaller wet biases over eastern Australia, and smaller dry biases elsewhere, except for Australia's tropical north. This marked reduction in wet bias magnitudes is a plausible contributing cause for the reduction in maximum temperature cold bias for the NARCliM2.0 RCMs. The CMIP6 and CMIP5 GCMs used to drive NARCliM2.0 and 1.5 RCMs generally show similar magnitudes of maximum temperature cold bias. This suggests that the underlying nature of the CMIP6 driving data is not a principal factor underlying the observed improvements for NARCliM2.0's simulation of maximum temperature. In fact, the RCMs appear to have a substantial influence on the reduced maximum temperature biases.

That NARCliM2.0 underestimates precipitation over tropical northern Australia during the wet season (summer) to a similar degree of magnitude to the NARCliM1.5 RCMs indicates that the newer models still struggle to accurately capture the strength of the Australian monsoon. However, whereas NARCliM1.x strongly overestimates precipitation over south-eastern and southern Australia, wet biases over these regions are reduced for NARCliM2.0 RCMs. This indicates that the newer models may confer an improved simulation of broad-scale processes associated with synoptic-scale systems interacting with the extratropical storm track over Australia (Grose et al., 2019).

NARCliM2.0 RCMs overestimate minimum temperatures across Australia, and these biases are larger relative to NARCliM1.5 but comparable to those of NARCliM1.0. The CMIP6 GCMs used to force NARCliM2.0 show substantially stronger warm biases for minimum temperature than the CMIP5 GCMs used for NARCliM1.5. This suggests that the increased warm bias for minimum temperature in NARCliM2.0-RCMs is partially inherited from the driving GCMs. However, as noted above, the Noah-MP LSM simulation of factors such as LAI and other aspects of vegetation as well as surface albedo in arid areas may contribute to the biases shown by the NARCliM2.0 RCMs. Moreover, the NARCliM2.0 ensemble mean reduces the overall minimum temperature bias of the CMIP6 GCM ensemble by almost half, attesting to the added value conferred by the NARCliM2.0 RCMs with respect to near-surface temperature variables.





In terms of NARCliM2.0 future climate projections, major changes between NARCliM genera-
tions such as differences in GHG scenarios mean that NARCliM2.0 projected temperature changes
differ in some respects to those of its predecessors. Overall, as is expected, projected warming is less
intense in NARCliM2.0 under SSP3-7.0 than for NARCliM1.5 under RCP8.5. Other differences in
the projections between NARCliM generations require further investigation in order to explain, such
as NARCliM1.5's latitudinal warming gradient for maximum temperature that contrasts with NAR-
CliM2.0's band of faster warming over central Australia relative to northern and southern regions.
Irrespective of these differences, all three NARCliM ensembles show statistically significant-agreeing
results for warming projections.
Precipitation projections for the different NARCliM generations show some key similarities,
such as reductions in mean annual precipitation over eastern Australia for NARCliM2.0 and NAR-
CliM1.5, though a difference is that these are statistically significant only for NARCliM1.5. The
NARCliM2.0-SSP3-7.0 and SSP1-2.6 ensembles differ in that the former generally projects wet
changes over northern and western Australia, whereas the latter is generally dry, results that appear
partially traceable to the underlying driving CMIP6-SSP data. Other notable differences are that some
NARCliM2.0 RCMs produce very similar precipitation projections for certain GCM-RCM combina-
tions, such as for ACCESS-ESM-1-5 forced R3 versus R5 under SSP3-7.0 (i.e. widespread dry pro-
jections for both RCMs). Conversely, in other instances, there are marked divergences between the R3
versus R5 precipitation projections when forced with the same GCM, for instance, UK-ESM-1-0-LL
under SSP3-7.0 where R3 projects stronger precipitation increases that are more geographically wide-
spread relative to R5. This raises the question of varying sources of uncertainty in the climate projec-
tions, i.e. to what extent these are attributable to GCMs versus RCMs, as well as other factors.
In summary, the CORDEX-CMIP6 NARCliM2.0 regional climate projections are a 10-member
ensemble comprising two configurations of the WRF RCM dynamically downscaling five GCMs un-
der three SSPs at 20 km resolution over CORDEX-Australasia and at 4 km convection-permitting
resolution over south-east Australia. The main aims of this manuscript are to describe the new
CORDEX-CMIP6 NARCliM2.0 RCM ensemble, explaining how and why its design choices were
made including the model test and evaluation approaches underlying these design decisions; and char-
acterise improvements in model skill in simulating the recent Australian climate relative to previous
generations of NARCliM, as well as differences in future climate projections. In addition to several
high-level model design changes, e.g. increased spatial resolution, a large (N=78) RCM-physics test
suite is evaluated to select two new WRF RCM configurations for CMIP6-forced NARCliM2.0 cli-
mate projections. Due to resource constraints and the aim to test a large number of RCM physics pa-
rameterisations, the NARCliM2.0 physics tests are performed for a single year. This is one reason
why the final selection of two production-grade RCMs for the CMIP6-NARCliM2.0 runs is based on
the CORDEX ERA5-forced 42-year evaluation simulations. The NARCliM2.0 physics tests identified
RCM configurations that generally performed well in simulating the recent Australian climate over

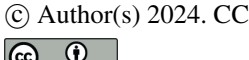



southeast Australia. A key finding was that WRF RCMs using the Noah-MP LSM generally out-
performed RCMs using other WRF LSMs in representing regional climate. Despite the performance
gains evident for RCMs using Noah-MP, RCM skill was superior over the temperate/coastal regions
of southeast Australia, relative to the semi-arid interior. These performance characteristics might be
linked to Noah-MP's development being focused on Northern Hemisphere mid-latitudes, including
assumptions such as accounting for differences in seasonality in the Northern versus Southern Hemi-
spheres by shifting the Northern Hemisphere LAI profiles by 6 months. For the southeast Australian
context, noting its distinctive coastal dry-sclerophyll and expansive inland grassland biomes, such
assumptions might lead to discontinuities in quantities such as LAI. Hence, future investigation into
processes such as land-surface coupling in NARCliM2.0 RCMs is warranted.
Overall, the CMIP6-NARCliM2.0 ensemble produces a good representation of recent mean cli-
mate that in several key respects improves upon the model skill of earlier NARCliM generations. This
study provides a foundation for more detailed investigations of the model biases and future climate
changes described here, including process-focused studies exploring their mechanisms. CORDEX-
CMIP6 NARCliM2.0 RCM data provide valuable resources to investigate projected climate changes,
their impacts on societies and natural systems, and potential climate change mitigation and adaptation
actions for the CORDEX-Australasia region.

## 9. Code Availability

The Weather Research and Forecasting (WRF) version 4.1.2 used in this study is freely available
from: https://github.com/coecms/WRF/tree/V4.1.2. A static copy of all scripts used for this study can
be found at: https://bitbucket.org/oehcas/narclim2-
0_design_and_evaluation_2024_support_materials/src/main/

## 10. Data Availability

Data for the NARCliM2.0 CMIP6-forced R3 and R5 RCMs are being made available via National
Computing Infrastructure (NCI). WRF namelist settings for the NARCliM2.0 CMIP6-forced R3 and
R5 RCMs are shown in Supplementary Material Figure S1 and are also available at:
https://bitbucket.org/oehcas/narclim2-0_design_and_evaluation_2024_support_materials/src/main/.
Data NARCliM1.5 RCMs are available via the New South Wales Climate Data Portal and CORDEX-
DKRZ. Data for NARCliM1.0 RCMs are available via the New South Wales Climate Data Portal.
CMIP6 GCM data are available via the Earth System Grid Federation.



## 11. Author Contribution

GDV and JPE designed the models and the simulations. FJ, ET, and CT setup the models and conducted the model simulations with contributions from JPE, JK, JA, DC, CR, SW, YL, MER, RG and JL. GDV prepared the manuscript with contributions from all co-authors.

## 12. Competing Interests

The authors declare that they have no conflict of interest, noting that JK is a Topic Editor of Geoscientific Model Development.

## 13. Funding

This research was supported by the New South Wales Department of Climate Change, Energy, the Environment and Water as part of the NARCliM2.0 dynamical downscaling project contributing to CORDEX Australasia. Funding was provided by the NSW Climate Change Fund for NSW and Australia Regional Climate Modelling (NARCliM) Project. This research was undertaken with the assistance of resources and services from the National Computational Infrastructure (NCI), which is supported by the Australian Government.

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
