# Peer review of "Design, evaluation and future projections of the NARCliM2.0 CORDEX-CMIP6"

_Geoscientific Model Development, 2024_

## Author Comment (AC1)

**Paper #**GMD-2024-87 | Model experiment description paper**: 'Design, evaluation and future projections of the NARCliM2.0 CORDEX-CMIP6 Australasia regional climate ensemble'**

**Author Comments (ACs) – Referee 1**

**Table 1. Anonymous Referee 1 (RC1) Comments**

| # | Issue Description | Discussion | Revision (in re-submitted manuscript) |
|---|---|---|---|
| | **Referee #1: General Comments** | | |
| 1 | The authors have compared the experimental designs and results across three generations of NARCliM RCMs. The latest iteration, NARCliM 2.0, features enhanced spatial resolution and utilizes CMIP6 experiment outputs as large-scale forcing data, representing advancements over earlier phases. The ensemble simulations of NARCliM 2.0 were conducted after a rigorous evaluation and selection process involving CMIP6 models and various physics configurations of the WRF model. This approach has the potential to provide more robust projections of regional climate over Australia. The ensemble simulations, incorporating diverse GCM-RCM combinations, make significant contributions to CORDEX. Therefore, I recommend acceptance pending minor revisions, including clarifications, correction, and reorganization in certain sections. Specific comments are outlined below: | We are very grateful to the reviewer for reviewing our work, for their positive remarks on this work and manuscript, and for recommending acceptance following Minor Revisions. | Please see point-by-point responses below. |
| | **Referee #1: Specific comments** | | |

| 2 | L108: Please replace "NARCliM2.0" with "NARCliM 2.0 (NARCliM 1.5)". | Agreed. | The naming of NARCliM is changed throughout the revised manuscript as suggested by the reviewer. |
|---|---|---|---|
| 3 | Section 3.2.1: It is unclear which variables were evaluated to assess CMIP6 GCM performance. Note that precipitation, daily maximum and minimum surface air temperatures do not serve as boundary conditions for driving the RCM. It would be preferable to evaluate U, V, T, Q, Z, SST, PSL for dynamical downscaling purposes. This issue should be properly addressed or discussed. | In this study, we evaluated the performance of CMIP6 GCMs by analysing mean climate, including annual and seasonal climatology of maximum and minimum temperatures, and precipitation; climate extremes, such as the 99th percentiles of daily maximum temperature and precipitation, and the 1st percentile of minimum temperature; as well as the teleconnections of ENSO, IOD, SAM, and their influence on Australian regional rainfall. The focus on temperature and rainfall is due to them being the best observed climate variables that provide the most direct comparison to observations (i.e. being gridded observational products). However, we also acknowledge the reviewer's suggestion of using variables such as U, V, T, Q, Z, SST, and PSL, which serve as initial and boundary conditions for driving the RCMs. If we want to evaluate U, V, T, Q, Z, PSL, etc, we would have to use re-analysis as the "surrogate truth/observations". This would be a useful thing to do, i.e., comparing CMIP6 against re-analysis for these variables, but it's a different exercise. These variables / this approach will be incorporated in future studies, and this is acknowledged in a revised version of the manuscript (please see text right). | The revised main text includes the statements below explaining the benefits of focusing on gridded observations of temperature and rainfall in the GCM evaluation, as well as acknowledging the reviewer's suggestion that variables such as U, V, T, Q, etc can be included in future GCM evaluation studies:

1) "Temperature and precipitation variables are chosen for evaluation because they are well-represented in high-quality gridded observational data sets for the Australian continent (King et al. 2013)."

2) "Variables such as winds (U, V), air temperature (T), water mixing ratio (Q), geopotential height (Z), sea surface temperature (SST), and sea level pressure (PSL) could be incorporated into future GCM evaluation studies as these variables serve as boundary conditions for driving RCMs. Evaluating such variables would require use of re-analysis data as surrogate observations." |
| 4 | Table 2: Please clarify how many GCM-RCM runs were conducted for CORDEX-CMIP6 NARCliM 2.0. Specify the combinations used. Were all five GCMs downscaled by seven RCMs each? Presenting this | The CORDEX-CMIP6 NARCliM 2.0 regional climate projections are a 10-member ensemble comprising two configurations of the WRF RCM dynamically downscaling the five shortlisted GCMs under three SSPs for 20 km and 4 km (i.e. | Text revised as follows:

1. The text preceding / introducing Table 2 is now revised to add mention that the five CMIP6 GCMs are used to force two, |

| | | | |
|---|---|---|---|
| | information in a table format would aid readers in quickly accessing these details. | convection-permitting scale). Although statements of this sort had been made at various points in the text of the submitted manuscript (please see example below), we agree with the reviewer that this key point can be further clarified (see changes in revised manuscript in column right).

The tremendous compute (financial) requirement to run these simulations necessitated us to be selective in the number of RCM configurations chosen to dynamically downscale the shortlisted CMIP6 GCMs. For instance, the ultimate outcome of the CORDEX ERA5-forced NARCliM 2.0 simulations and their evaluation was the selection of the two definitive RCM configurations R3 and R5 to run the CMIP6-forced phase of NARCliM 2.0.
An example of existing text in submitted manuscript (see lines 898-901); added text in revised manuscript shown in column right:

"In summary, the CORDEX-CMIP6 NARCliM 2.0 regional climate projections are a 10-member ensemble comprising two configurations of the WRF RCM dynamically downscaling five GCMs under three SSPs at 20 km resolution over CORDEX-Australasia and at 4 km convection-permitting resolution over south-east Australia" | definitive RCMs comprising NARCliM 2.0 CORDEX-CMIP6:

"As a result of the above process, the five CMIP6 GCMs listed in Table 2 are selected to force each of the two definitive NARCliM 2.0 RCMs selected via the RCM physics testing and ERA5 evaluation processes."

2. The caption for Table 2 is also revised accordingly:

"**Table 2.** Basic details of the CMIP6 GCMs used to force two RCMs comprising the NARCliM 2.0 CORDEX-CMIP6 ensemble." |
| 5 | L423-424: The authors employed a cold restart for the SSP experiments. Did the authors examined the duration required for deep soil spin-up? Why not use soil moisture from a historical RCM run in 2014 | Ideally, we would complete the long-term historical simulation first and use the final restart file from this simulation to initialize the first SSP simulation. However, due to time constraints we had to run historical and SSP simulations | Following text added to the revised manuscript: |

| | | concurrently, using a one-year spin-up period. In this study, we conducted a cold restart for the historical simulation in 2014 and used the final restart files from 2014 to initialize the first SSP simulation in 2015. We also evaluated the time needed for deep soil spin-up, which is approximately 3 to 6 months for different Australia regions. To account for this, we used a 12-month spin-up period, which is sufficient to minimize the impact of the cold restart. | "We tested the time duration required for soil moisture to equilibrate from the cold start and found that 1 year is sufficient." |
|---|---|---|---|
| or ERA5 reanalysis as initial conditions for the SSP experiments? | | | |
| 6 | Section 4 Evaluation methods: these evaluation methods were already used in previous sections. It would improve clarity to present this section earlier in the manuscript. | Thanks for this suggestion: we agree it is better to swap section 3 and 4 and make some changes accordingly. | Main text to be revised as suggested. |
| 7 | L453-456: RMSE and PSS are typically used to assess model performance in simulating individual variables. However, it remains unclear how overall RCM performance in simulating multiple variables is determined. Did the authors normalized the biases/RMSEs when sum them together? Otherwise, the biases/RMSEs are in different order of magnitude. The authors may consider employing the Model Climate Performance Index (Gleckler et al., 2008) or multivariable integrated skill score (Zhang et al., 2021) for a comprehensive assessment in terms of the model performance in simulating multiple variables. | There are several methods to evaluate the overall performance of RCMs. In this study, we ranked the RCMs individually based on their bias, RMSE, and PSS for maximum temperature, minimum temperature, and precipitation. Each variable was ranked separately for each metric. The ranks were then summed to determine the overall ranking for each RCM. Thank you for suggesting these references; in particular, in future studies we will try the approach of Zhang et al., (2021). | Text below added to the revised manuscript to provide more clarity on this matter:

"There are several methods to evaluate the overall performance of RCMs. In this study, we ranked the RCMs individually based on their bias, RMSE, and PSS for maximum temperature, minimum temperature, and precipitation. Each variable was ranked separately for each metric. The ranks were then summed to determine the overall ranking for each RCM." |
| 8 | L699: Please replace "CMPI6" with "CMIP6". | Thank you for pointing that out – corrected. | "CMPI6" corrected to "CMIP6". |
| 9 | L707-712: Could you explain why projected changes in TAS exhibit distinct spatial patterns between NARCliM 2.0 and NARCliM 1.5/1.0? | Thanks for this comment. In this work, we looked at future projections of mean maximum temperature (TASMAX) rather than mean temperature (TAS). Given your comment, we compared differences in the spatial patterns of projected changes in both TAS and TASMAX | The manuscript had stated the need for further work in this space, noting our comments in the column left.

Lines 913-916: |

| | | between CMIP6 and CMIP5 GCMs (see Figure 1 below this table). Both GCM generations show broadly similar spatial patterns of change (at least qualitatively). However, there are clear differences in magnitude, e.g. whilst both CMIP5-6 show stronger warming changes across an east-west band of central Australia, the magnitude of change is larger for CMIP5, probably in large part to the differences in GHG assumptions (See Figure 1 below this table). Additionally, GCM skill in simulating observed TASMAX is fairly similar for both GCM generations (see Supporting Information Figure S7), noting though that the spatial patterns of bias are somewhat different (e.g. the CMIP6 ensemble mean is more cold biased over northern Australia than CMIP5; conversely the CMIP5 GCM ensemble mean is more cold biased over southern and eastern Australia).

This topic requires an additional in-depth investigation to understand and explain which is out of scope for this paper. For example, TMAX is usually driven at the larger scale by changes in MSLP, e.g, the sub-tropical ridge and its intensification, this in turn probably affects changes in precip. and surface energy balance, so we would need to examine changes in potentially: MSLP, precip., soil moisture, sensible, latent heat fluxes etc. Our aim with this current work is to explain key model design processes and the basic performance characteristics of the NARCliM models, i.e. to lay a foundation for future work in this space. There might be several | "Other differences in the projections between NARCliM generations require further investigation in order to explain, such as NARCliM 1.5's latitudinal warming gradient for maximum temperature that contrasts with NARCliM 2.0's band of faster warming over central Australia relative to northern and southern regions." |

| | | factors that underlie the different/distinct spatial patterns in projected temperature changes for NARCliM 2.0 and NARCliM 1.x. For instance, changes in model spatial resolution are one possible candidate, given that the resolution of CMIP6 GCMs is higher than CMIP5 GCMs, and the same applies to NARCliM 2.0 RCMs versus its predecessors. However, we expect that there will be other factors that explain the observed differences in NARCliM RCM behaviour. | |
|---|---|---|---|
| 10 | Fig.15: The quality of this figure appears low. Why do the stippling areas form very regular circles in the many subpanels, e.g., b, c, e, n, p, t, u, v? Consider presenting these figures as supplementary material and summarizing the statistics using a Taylor diagram. | We agree that the quality of the original Figure 15 was insufficient: this figure is now revised, e.g. with DPI increased from 300 to 600, stippling size increased, panel title font size increased, etc., – please see revised figure below this table. | Figure 15 revised (please see example below this table). |
| 11 | L804-816: These discussions are somewhat tangential to the study's main focus and could be shortened or omitted. Instead, further investigate/discuss the differences in projected changes in the surface air temperature and precipitation among the three generations of NARCliM. For example, explore why widespread wet biases observed in NARCliM 1.x are substantially reduced in NARCliM 2. Are these biases attributable to GCMs, RCMs, or both? | This study focuses on summarizing the improvements in the NARCliM2.0 design, including the incorporation of the Noah-MP land surface model, which has significantly reduced cold biases in both ERA5 and GCM-driven simulations. This section discusses the successful application of Noah-MP in other regions, which aligns with the results we achieved in our project. Additionally, we explore how Noah-MP performance in Australia can be further enhanced by selecting specific settings rather than relying on default ones for future regional climate modelling. We believe these discussions are relevant to the focus of the study. We also appreciate the reviewer's suggestion to address why the wet biases in NARCliM1.0 and NARCliM1.5 were reduced in NARCliM2.0. The | Revised text now includes the following additional text (based on that in column left): "Overall, the CMIP6 GCMs used to drive NARCliM 2.0 show marginally reduced wet biases relative to the CMIP5 GCMs used to drive NARCliM1.5 RCMs (e.g. area-averaged ensemble mean absolute biases are 7.13 mm and 8.89 mm, respectively; Supporting Information Figure S15). This suggests that the underlying nature of the CMIP6 driving data is not the principal factor underlying the observed improvements for NARCliM 2.0's simulation of mean precipitation. In fact, the RCMs appear to have a substantial influence on the reduced maximum |

main aims of the present paper are more focused on introducing the model design processes, and the basic performance profiles of the new models as compared to the previous generations, with more detailed explorations explaining differences in model skill etc to be the topics of future work. For instance, this topic is also being discussed in more detail in another paper, 'Three Generations of NARCliM: Model Evaluation and Future Projections over CORDEX Australia,' which is currently under review.

That said, we can suggest initial explanations as to why widespread wet biases observed in NARCliM 1.x are substantially reduced in NARCliM 2.0:

Overall, the CMIP6 GCMs used to drive NARCliM 2.0 show marginally reduced wet biases relative to the CMIP5 GCMs used to drive NARCliM1.5 RCMs (e.g. area-averaged ensemble mean absolute biases are 7.13 mm and 8.89 mm, respectively; Supporting Information Figure S15). This suggests that the underlying nature of the CMIP6 driving data is not the principal factor underlying the observed improvements for NARCliM 2.0's simulation of mean precipitation. In fact, the RCMs appear to have a substantial influence on the reduced maximum temperature biases. Conversely, in terms of RCM design features, the use of the Noah-MP LSM in the NARCliM 2.0 RCM physics tests conferred overall RCM skill improvements relative to RCMs using the Noah-Unified LSM for both mean max

temperature biases. Conversely, in terms of RCM design features, the use of the Noah-MP LSM in the NARCliM 2.0 RCM physics tests con-ferred overall RCM skill improvements relative to RCMs using the Noah-Unified LSM for both mean max temperature and precipitation. The developers of Noah-MP suggest that some limitations in the Noah-Unified LSM have been modified to better represent several parameters such as soil moisture and heat fluxes, leaf area-rainfall interaction, vegetation and canopy temperature distinction, drainage of soil, and runoff. The production NARCliM2.0 RCMs forced with CMIP6 GCMs used Noah-MP, whereas NARCliM1.x RCMs used Noah-Unified. Given these performance improvements observed for RCMs using Noah-MP versus RCMs using Noah-Unified, it's plausible that the different land surface schemes (i.e. Noah-MP for NARCliM 2.0 versus Noah-Unified for NARCliM 1.x) play a role in the improved NARCliM2.0 RCM skill in simulating mean precipitation (as well as max temp), for instance, via changing the land surface feedback (via soil moisture) to the simulation of precipitation. However, this possibility requires more extensive investigation via future studies."

| | | temperature and precipitation. The developers of Noah-MP suggest that some limitations in the Noah-Unified LSM have been modified to better represent several parameters such as soil moisture and heat fluxes, leaf area-rainfall interaction, vegetation and canopy temperature distinction, drainage of soil, and runoff. The production NARCliM2.0 RCMs forced with CMIP6 GCMs used Noah-MP, whereas NARCliM1.x RCMs used Noah-Unified. Given these performance improvements observed for RCMs using Noah-MP versus RCMs using Noah-Unified, it's plausible that the different land surface schemes (i.e. Noah-MP for NARCliM 2.0 versus Noah-Unified for NARCliM 1.x) play a role in the improved NARCliM2.0 RCM skill in simulating mean precipitation (as well as max temp), for instance, via changing the land surface feedback (via soil moisture) to the simulation of precipitation. However, this possibility requires more extensive investigation via future studies. | |
|---|---|---|---|

[Figure]

**Figure 1 (New)**. Future projections of mean maximum temperature for the ensemble means of CMIP5 GCMs forcing NARCliM 1.5 and CMIP6 GCMs forcing NARCliM 2.0

[Figure]

**Figure 15**: revised version

---

## Author Comment (AC3)

**Paper #**GMD-2024-87 | Model experiment description paper**: 'Design, evaluation and future projections of the NARCliM2.0 CORDEX-CMIP6 Australasia regional climate ensemble'**

**Author Comments (ACs) – Referee 3**

**Table 3. Anonymous Referee 3 (RC3) Comments**

| # | Issue Description | Discussion | Revision (in re-submitted manuscript) |
|---|---|---|---|
| | **Referee #3: General Comments** | | |
| 1 | The authors present the regional climate model NARCliM2.0 and evaluate it using various GCM and RCM ensembles, as well as its precursor versions 1.0 and 1.x. The research topic is highly interesting, and the research work has been conducted meticulously and comprehensively, making it very valuable for regional climate model evaluation and future climate projections in Australia. The research framework is also inspiring for regional climate science, particularly for other regions with large populations. The manuscript is well-written and well-structured. In conclusion, I recommend publication in GMD after the specific comments listed below have been addressed. | We thank the reviewer for reviewing this manuscript, for the positive and constructive remarks on our work, and for recommending publication after addressing your specific comments below. | |
| | **Referee #3: Specific comments** | | |
| 2 | Line 81: "and 3) summarise the climate projections produced by CMIP6-NARCliM2.0 and how these" to "3) summarise the climate projections produced by CMIP6-NARCliM2.0 and how these". | Thanks for pointing this out – text changed as suggested. | Text revised. |
| 3 | Line 83-88: "section x." to "Section x". Please check all "section x.x" and "sect. x.x" in the manuscript. | Agreed. | Text revised as suggested. |
| 4 | Line 108-109: "NARCliM2.0 RCMs have a 20 km resolution CORDEX-Australasia domain (versus 50 km) and 4 km (versus 10 km) domain over southeast Australia and use 45 (versus | There is no strict requirement for vertical resolution to match horizontal resolution. However, in NARCliM 2.0, | No change in the main text. |

| | | | |
|---|---|---|---|
| | 30) vertical levels". The horizontal resolution in NARCLiM2.0 has more than doubled resolutions, yet the vertical resolution is from 30 to 45 vertical levels. What do authors think of the choice of 45 levels instead 60 or even more? | we carefully balanced the horizontal and vertical resolutions. By increasing the number of vertical levels from 30 to 45, we primarily enhanced the vertical resolution within the boundary layer, allowing for a better representation of vertical profiles of temperature, moisture, and winds. The vertical grid spacing in the boundary layer is around 50–200 meters, which is sufficient to resolve important vertical processes. In early testing for NARCliM2.0, we also tested using 60 and 75 vertical levels. The surface climate produced was very similar to when using 45 levels, but the computational cost was substantially larger. Given that finding and resource constraints, we determined that 45 vertical levels could effectively meet the objectives of NARCliM 2.0. | |
| 5 | Line 142: "manuscripts describe elements shown in Figure 2, and which are therefore only summarised briefly in", remove "and". | Agreed: 'and' not needed. | Text revised as suggested. |
| 6 | Line 164-167: "The performances of the different test RCM configurations are evaluated, ultimately selecting a subset of seven RCMs for subsequent downscaling of ERA5 reanalysis and comprising the CORDEX evaluation experiment." To "The performance of the different test RCM configurations is evaluated, ultimately leading to the selection of a subset of seven RCMs for subsequent downscaling of ERA5 reanalysis as part of the CORDEX evaluation experiment". | Agreed, text revised as suggested. | Text revised. |
| 7 | Line 170: 'production' should be "production". Please check all 'something' in the manuscript. | In the revised manuscript, we have avoided the use of text like 'production' | Text revised as follows: |

| | | – production is sufficient, hence quotes removed as per example right. | "Evaluating these ERA5-forced simulations informs selection of two production RCMs for CMIP6-forced downscaling" |
|---|---|---|---|
| 8 | Line 190-191: "Non-normally distributed variables (e.g. snow depth and precipitation) are checked for global minima and maxima only." To "Non-normally distributed variables (e.g., snow depth and precipitation) are checked only for global minima and maxima." Please check all "e.g." in the manuscript. | Text in sentence corrected as suggested, including correction for all "e.g." as suggested. | Text revised. |
| 9 | Line 201: "Check that changes over time are within realistic ranges (i.e. assess temporal gradients)." To "Check that changes over time are within realistic ranges (i.e., assess temporal gradients)." Please check all "i.e." in the manuscript. | Text changed as suggested. | Text revised throughout. |
| 10 | Line 354-355: "Some studies have shown using this option improves modelling of soil moisture (e.g. Zhuo et al., 2019)." to "Some studies have shown **that** using this option improves **the** modeling of soil moisture (**e.g.,** Zhuo et al., 2019)." | Thanks – changes implemented as suggested. | Manuscript text revised. |
| 11 | Table 9: I am confused about how exactly the "R1-R7" RCMs are shortlisted. It said in Line 609 that "RCMs are shortlisted from the set of 20 if they rank highly for both performance and independence", but it is not clear how the RCMs are ranked from "R1" to "R7". Please explain it in more detail. | We shortlisted the 7 RCMs from the shortlisted 20 candidates based on their performance and independence ranking. However, there was no ranking from R1 to R7 per se, this is just a naming convention chosen at the point of embarking on the next stage of the design/model evaluation process with was the ERA5-forced RCM simulations conducted for the CORDEX ERA5 evaluations. Only after completing these CORDEX ERA5 evaluations did we compare the performance of R1-R7 and at that point we selected R3 and R5 for the subsequent CMIP6-forced RCM simulations – please see Di Virgilio et | We have added the following note to the revised main text:

"We note here that R1-R7 are simply a chronological naming convection and do not imply any ranking." |

| | | al.,([https://gmd.copernicus.org/preprints/gmd-2024-41/gmd-2024-41.pdf](https://gmd.copernicus.org/preprints/gmd-2024-41/gmd-2024-41.pdf)) for further details. | |
|---|---|---|---|
| 12 | Figure 15: there are many subfigures and their titles are not easy to read. Please consider improve the visualization. | Agreed, the original Figure 15 was of insufficient quality (e.g. 300 DPI), so we have increased to 600 DPI and improved clarity of stippling and titles as far as is possible for a figure with 31 individual plot panels. | Figure 15 revised, please see example below this table. |
| 13 | Line 777: "with 4 RCMs using BMJ, 2 RCMs using Tiedtke, and 1 using Kain-Fritsch." Please give the references to the cumulus parameterisations. | This is a good idea – we have included references for all physics used in the study. | References for all physics settings used added into Table 3 in the revised manuscript. |

[Figure]

**Figure 15**: revised version

---

## Author Response (AR1)

**Paper #GMD-2024-87 | Model experiment description paper:** 'Design, evaluation and future projections of the NARCliM2.0 CORDEX-CMIP6 Australasia regional climate ensemble'

Dear Prof. Rahimi-Esfarjani,

We thank the Editor and the three Referees for their constructive input, and for assessing this manuscript as suitable to publication following the opportunity to implement revisions.

As you will see from our point-by-point responses in Tables 1-3 below, we have carefully gone through all the Referee comments and suggestions.

We believe that the Referee comments have helped strengthen this manuscript, and we are very grateful for their reviews.

Kind regards,

Giovanni Di Virgilio, Fei Ji, Eugene Tam, Jason Evans, Jatin Kala, Julia Andrys, Christopher Thomas, Dipayan Choudhury, Carlos Rocha, Stephen White, Yue Li, Moutassem El Rafei, Rishav Goyal, Matthew Riley, Jyothi Lingala

**Table 1. Anonymous Referee 1 (RC1) Comments**

| # | Issue Description | Discussion | Revision (in re-submitted manuscript) |
|---|---|---|---|
| | **Referee #1: General Comments** | | |
| 1 | The authors have compared the experimental designs and results across three generations of NARCliM RCMs. The latest iteration, NARCliM 2.0, features enhanced spatial resolution and utilizes CMIP6 experiment outputs as large-scale forcing data, representing advancements over earlier phases. The ensemble simulations of NARCliM 2.0 were conducted after a rigorous evaluation and selection process involving CMIP6 models and various physics configurations of the WRF model. This approach has the potential to provide more robust projections of regional climate over Australia. The ensemble simulations, incorporating diverse GCM-RCM combinations, make significant contributions to CORDEX. Therefore, I recommend acceptance pending minor revisions, including clarifications, correction, and reorganization in certain sections. Specific comments are outlined below: | We are very grateful to the referee for their review, for their positive remarks on this work and manuscript, and for recommending acceptance following Minor Revisions. | Please see point-by-point responses below. |
| | **Referee #1: Specific comments** | | |
| 2 | L108: Please replace "NARCliM2.0" with "NARCliM 2.0 (NARCliM 1.5)". | Agreed. | The naming of NARCliM is changed throughout the revised manuscript as suggested by the reviewer. |
| 3 | Section 3.2.1: It is unclear which variables were evaluated to assess CMIP6 GCM performance. Note that precipitation, daily maximum and minimum surface air temperatures do not serve as boundary conditions for driving the RCM. It would be preferable to evaluate U, V, T, Q, Z, SST, PSL for dynamical downscaling purposes. This issue should be properly addressed or discussed. | In this study, we evaluated the performance of CMIP6 GCMs by analysing mean climate, including annual and seasonal climatology of maximum and minimum temperatures, and precipitation; climate extremes, such as the 99th percentiles of daily maximum temperature and precipitation, and the 1st percentile of minimum temperature; as well as the teleconnections of | As suggested by the reviewer, this issue is now discussed in the revised manuscript. The revised main text includes the statements below explaining the benefits of focusing on gridded observations of temperature and rainfall in the GCM evaluation, as well as acknowledging the reviewer's suggestion that variables such |

| | | |
|---|---|---|
| | ENSO, IOD, SAM, and their influence on Australian regional rainfall. The focus on temperature and rainfall is due to them being the best observed climate variables that provide the most direct comparison to observations (i.e. being gridded observational products). However, we also acknowledge the reviewer's suggestion of using variables such as U, V, T, Q, Z, SST, and PSL, which serve as initial and boundary conditions for driving the RCMs. If we want to evaluate U, V, T, Q, Z, PSL, etc, we would have to use re-analysis as the "surrogate truth/observations". This would be a useful thing to do, i.e., comparing CMIP6 against re-analysis for these variables, but it is a different exercise. These variables / this approach will be incorporated in future studies, and **this is acknowledged in a revised version of the manuscript (please see text right).** | as U, V, T, Q, etc are appropriate for inclusion in future GCM evaluation studies (lines 323-336):

"We evaluated the performances of individual CMIP6 GCMs in simulating the following aspects of the observed historical climate of Australia:

▪ annual and seasonal climatologies and daily distributions of maximum and minimum temperatures and precipitation;
▪ climate extremes, such as the 99th percentiles of daily maximum temperature and precipitation, and the 1st percentile of minimum temperature;
▪ teleconnections of climate modes and Australian regional rainfall.

Temperature and precipitation variables are chosen for evaluation because, being well-represented in high-quality gridded observational data sets for the Australian continent, they provide the most direct comparison to observations (King et al., 2013). They are also often prioritised for impact studies. Given variables such as winds (U, V), air temperature (T), water mixing ratio (Q), geopotential height (Z), sea surface temperature (SST), and sea level pressure (PSL) serve as boundary conditions for driving RCMs, these could be incorporated into future GCM evaluation studies. However, evaluating such variables |

| | | | would require use of re-analysis data as surrogate observations." |
|---|---|---|---|
| 4 | Table 2: Please clarify how many GCM-RCM runs were conducted for CORDEX-CMIP6 NARCliM 2.0. Specify the combinations used. Were all five GCMs downscaled by seven RCMs each? Presenting this information in a table format would aid readers in quickly accessing these details. | The CORDEX-CMIP6 NARCliM 2.0 regional climate projections are a 10-member ensemble comprising two configurations of the WRF RCM dynamically downscaling the five shortlisted GCMs under three SSPs for 20 km and 4 km (i.e. convection-permitting scale). Although statements of this sort had been made at various points in the text of the submitted manuscript (please see example below), we agree with the reviewer that this key point can be further clarified (see changes in revised manuscript in column right).

The tremendous compute (financial) requirement to run these simulations necessitated us to be selective in the number of RCM configurations chosen to dynamically downscale the shortlisted CMIP6 GCMs. For instance, the ultimate outcome of the CORDEX ERA5-forced NARCliM 2.0 simulations and their evaluation was the selection of the two definitive RCM configurations R3 and R5 to run the CMIP6-forced phase of NARCliM 2.0.

An example of existing text describing the ensemble composition in the original manuscript (see lines 898-901); and added text in revised manuscript shown in column right:

"In summary, the CORDEX-CMIP6 NARCliM 2.0 regional climate projections are a 10-member ensemble comprising two configurations of the | Text revised as follows (lines 364-368):

1. The text preceding / introducing Table 2 is now revised to add mention that the five CMIP6 GCMs are used to force two, definitive RCMs comprising NARCliM 2.0 CORDEX-CMIP6:

"As a result of the above process, the five CMIP6 GCMs listed in Table 2 are selected to force each of the two definitive NARCliM 2.0 RCMs selected via the RCM physics testing and ERA5 evaluation processes."

2. The caption for Table 2 is also revised accordingly:

"**Table 2.** Basic details of the CMIP6 GCMs used to force the two definitive RCMs comprising the NARCliM 2.0 CORDEX-CMIP6 ensemble." |

| | | | |
|---|---|---|---|
| | | WRF RCM dynamically downscaling five GCMs under three SSPs at 20 km resolution over CORDEX-Australasia and at 4 km convection-permitting resolution over south-east Australia" | |
| 5 | L423-424: The authors employed a cold restart for the SSP experiments. Did the authors examined the duration required for deep soil spin-up? Why not use soil moisture from a historical RCM run in 2014 or ERA5 reanalysis as initial conditions for the SSP experiments? | Ideally, we would complete the long-term historical simulation first and use the final restart file from this simulation to initialize the first SSP simulation. However, due to time constraints we had to run historical and SSP simulations concurrently, using a one-year spin-up period. In this study, we conducted a cold restart for the historical simulation in 2014 and used the final restart files from 2014 to initialize the first SSP simulation in 2015. We also evaluated the time needed for deep soil spin-up, which is approximately 3 to 6 months for different Australia regions. To account for this, we used a 12-month spin-up period, which is sufficient to minimize the impact of the cold restart. | Following new text in **bold** added to the revised manuscript (lines 522-25):

"A cold restart was performed on the last Historical experiment year (2014), thus enabling the SSP1-2.6 and SSP3-7.0 experiments to be run for 2015-2100 concurrently with the Historical experiment. **Testing the time duration required for soil moisture to equilibrate from the cold start showed that 1 year is sufficient**." |
| 6 | Section 4 Evaluation methods: these evaluation methods were already used in previous sections. It would improve clarity to present this section earlier in the manuscript. | Thanks for this suggestion: the original submission presented 'Section 3. NARCliM 2.0 design and production process overview' before it presented 'Section 4 Evaluation Methods'. We agree with your suggestion that in the revised manuscript it is better to swap the order of presentation of Section 3 and Section 4 and make some changes accordingly (e.g. re-numbering of these two sections). | Main text revised as suggested: 'Evaluation Methods' (renumbered to Section 3) now presented before 'NARCliM 2.0 design and production process overview' (which is now Section 4). |
| 7 | L453-456: RMSE and PSS are typically used to assess model performance in simulating individual variables. However, it remains unclear how overall RCM performance in simulating multiple variables is determined. Did the authors normalized the biases/RMSEs when sum them together? Otherwise, | There are several methods to evaluate the overall performance of RCMs. In this study, we ranked the RCMs individually based on their bias, RMSE, and PSS for maximum temperature, minimum temperature, and precipitation. Each variable was ranked separately for each metric. The ranks | Text below added to the revised manuscript to provide more clarity on this matter (lines 167-169):

"There are several methods to evaluate the overall performance of RCMs. In this study, |

| | | | |
|---|---|---|---|
| | the biases/RMSEs are in different order of magnitude. The authors may consider employing the Model Climate Performance Index (Gleckler et al., 2008) or multivariable integrated skill score (Zhang et al., 2021) for a comprehensive assessment in terms of the model performance in simulating multiple variables. | were then summed to determine the overall ranking for each RCM. Thank you for suggesting these references; in particular, in future studies we will try the approach of Zhang et al., (2021). | we ranked the RCMs individually based on their bias, RMSE, and PSS for maximum temperature, minimum temperature, and precipitation. Each variable was ranked separately for each metric. The ranks were then summed to determine the overall ranking for each RCM." |
| 8 | L699: Please replace "CMPI6" with "CMIP6". | Thank you for pointing that out – corrected. | "CMPI6" corrected to "CMIP6". |
| 9 | L707-712: Could you explain why projected changes in TAS exhibit distinct spatial patterns between NARCliM 2.0 and NARCliM 1.5/1.0? | Thanks for this comment. In this work, we looked at future projections of mean maximum temperature (TASMAX) rather than mean temperature (TAS). Given your comment, we compared differences in the spatial patterns of projected changes in both TAS and TASMAX between CMIP6 and CMIP5 GCMs (please see Figure 1 below this table, p.10; shown here, but not added to revised manuscript). Both GCM generations show broadly similar spatial patterns of change (at least qualitatively). However, there are clear differences in magnitude, e.g. whilst both CMIP5-6 show stronger warming changes across an east-west band of central Australia, the magnitude of change is larger for CMIP5, probably in large part to the differences in GHG assumptions. Additionally, GCM skill in simulating observed TASMAX is fairly similar for both GCM generations (see Supporting Information Figure S7), noting though that the spatial patterns of bias are somewhat different (e.g. the CMIP6 ensemble mean is more cold biased over northern Australia than CMIP5; conversely the CMIP5 GCM ensemble mean is more cold biased over southern and eastern Australia). | The manuscript had stated the need for further work in this space, noting our comments here in the column left.

Lines 928-931:

"Other differences in the projections between NARCliM generations require further investigation in order to explain, such as NARCliM 1.5's latitudinal warming gradient for maximum temperature that contrasts with NARCliM 2.0's band of faster warming over central Australia relative to northern and southern regions." |

| | | This topic requires an additional in-depth investigation to understand and explain. For example, TMAX is usually driven at the larger scale by changes in MSLP, e.g, the sub-tropical ridge and its intensification, this in turn probably affects changes in precip. and surface energy balance, so we would need to examine changes in potentially: MSLP, precip., soil moisture, sensible, latent heat fluxes etc. Our aim with this current work is to explain key model design processes and the basic performance characteristics of the NARCliM models, i.e. to lay a foundation for future work in this space, hence an investigation like the above, whilst very interesting, is more within the scope of a new study. There might be several factors that underlie the different/distinct spatial patterns in projected temperature changes for NARCliM 2.0 and NARCliM 1.x. For instance, changes in model spatial resolution are one possible candidate, given that the resolution of CMIP6 GCMs is higher than CMIP5 GCMs, and the same applies to NARCliM 2.0 RCMs versus its predecessors. However, we expect that there will be other factors that explain the observed differences in NARCliM RCM behaviour. | |
|---|---|---|---|
| 10 | Fig.15: The quality of this figure appears low. Why do the stippling areas form very regular circles in the many subpanels, e.g., b, c, e, n, p, t, u, v? Consider presenting these figures as supplementary material and summarizing the statistics using a Taylor diagram. | We agree that the quality of the original Figure 15 was insufficient: this figure is now revised, e.g. with DPI increased from 300 to 600, stippling size increased, panel title font size increased, etc., – please see revised figure below this table. | Figure 15 revised as described in column left and included in the revised manuscript (please also see revised figure below this table, p. 11). |
| 11 | L804-816: These discussions are somewhat tangential to the study's main focus and could be shortened or omitted. Instead, further | This study focuses on summarizing the improvements in the NARCliM2.0 design, including the incorporation of the Noah-MP land | As suggested, the section of text in question has been substantially shortened in the revised manuscript. Additionally, |

investigate/discuss the differences in projected changes in the surface air temperature and precipitation among the three generations of NARCliM. For example, explore why widespread wet biases observed in NARCliM 1.x are substantially reduced in NARCliM 2. Are these biases attributable to GCMs, RCMs, or both?

surface model, which has significantly reduced cold biases in both ERA5 and GCM-driven NARCliM2.0 simulations. Section 8.1 of the main text (near the start of the Discussion) discusses the application of Noah-MP by other studies. Additionally, we explore how Noah-MP performance in Australia can be further enhanced by selecting specific settings rather than relying on default ones for future regional climate modelling. Whilst we believe that, overall, several of these discussions are relevant to the focus of the study, we agree with your suggestion to shorten the text at lines 804-816. Hence, we have streamlined this section in the revised manuscript, removing the section of text starting 'In an assessment of the performances of several WRF-LSMs for Sub-Saharan Africa …' which was at lines 804 to 814 in the original text (text removed shown below this paragraph).

"In an assessment of the performances of several WRF-LSMs for Sub-Saharan Africa, Glotfelty et al. (2021) noted deficiencies in the simulation of land use and land cover change (LULCC) parameters such as surface albedo by Noah-MP. Despite these deficiencies, the spatial patterns and magnitudes of temperature and precipitation were well-represented by Noah-MP. However, the land surface parameter errors impacted the magnitude and sign of LULCC-induced changes in temperature and precipitation. These deficiencies were linked to substantial underestimations of surface albedo in arid areas due to inaccurate soil albedo treatments by Noah-MP. Moreover, errors

based on the reviewer's feedback, the revised text now includes the following additional text (please see Sect. 8.2, lines 873-888):

"The extent to which NARCliM2.0's improved simulation of precipitation might be attributable to its driving data warrants consideration. Overall, the CMIP6 GCMs used to drive NARCliM 2.0 show marginally reduced wet biases versus the CMIP5 GCMs used for NARCliM1.5 (e.g. area-averaged ensemble mean absolute biases are 7.13 mm and 8.89 mm, respectively; Supporting Information Figure S15). This suggests that the underlying nature of the CMIP6 driving data might not be the principal factor underlying the observed improvements for NARCliM 2.0's simulation of mean precipitation. Conversely, in terms of RCM design features, the use of the Noah-MP LSM in the NARCliM 2.0 RCM physics tests conferred overall RCM skill improvements relative to RCMs using the Noah-Unified LSM for both mean precipitation and mean maximum temperature. As noted above, the developers of Noah-MP suggest that some features of the Noah-Unified LSM have been modified to better represent several parameters. The production NARCliM2.0 RCMs used Noah-MP, whereas NARCliM1.x RCMs used Noah-Unified. Given these performance improvements

| | | in Noah-MP's LAI profiles may occur because it was developed principally for application in Northern Hemisphere mid-latitudes. It is possible that modifying/tuning Noah-MP to specific aspects of the Australian context would yield performance benefits for follow-up dynamical downscaling. Overall, these authors concluded that "Noah-MP is least flawed of the [WRF] default LSMs". | observed for RCMs using Noah-MP versus using Noah-Unified, it is plausible that the newer LSM contributes to the improved NARCliM2.0 skill in simulating precipitation and maximum temperature, for instance, via changing the land surface feedback (via soil moisture) to the simulation of precipitation. This possibility requires more extensive investigation via future studies." |
| | | We also appreciate the reviewer's suggestion to address why the wet biases in NARCliM1.0 and NARCliM1.5 were reduced in NARCliM2.0. The main aims of the present paper are more focused on introducing the model design processes, and the basic performance profiles of the new models as compared to the previous generations, with more detailed explorations explaining differences in model skill etc to be the topics of future work. | |
| | | That said, we can suggest initial explanations as to why widespread wet biases observed in NARCliM 1.x are substantially reduced in NARCliM 2.0: please see in column right new text added to the revised manuscript. | |

[Figure]

**Figure 1 (New, see comment #9 above)**. Future projections of mean maximum temperature for the ensemble means of CMIP5 GCMs forcing NARCliM 1.5 and CMIP6 GCMs forcing NARCliM 2.0

[Figure]

**Figure 15** revised figure and caption: "**Figure 15.** Climate change projections (1990-2009 versus 2060-2079) for annual mean precipitation for NARCliM ensemble mean climate change signals (a,l,s) and for individual ensemble members for each generation of NARCliM simulation (NARCliM 2.0 under SSP3-7.0, NARCliM 1.5 under RCP8.5 and NARCliM 1.0 under SRES A2). Significance stippling as per Figure 9."

**Table 2. Anonymous Referee 2 (RC2) Comments**

| # | Issue Description | Discussion | Revision (in re-submitted manuscript) |
|---|---|---|---|
| | **Referee #2: General Comments** | | |
| 1 | The authors perform extensive testing of WRF physics schemes for future regional climate projections over SE Australia. Impressively, the model is run at 4km convective permitting resolution. After choosing operational configurations, the authors document the historical biases and future projections. While the analysis is rather simple, it is very helpful that comparisons are made against previous generations of NARCLIM. I think this will form a very important foundational paper. I suggest major revisions based on my comments below, which mostly relate to clarifying important points and improving the presentation and interpretation of results. | We thank the reviewer for reviewing our manuscript and for their constructive comments on our work, including their view that this will form a very important foundational paper. Please see our responses to the reviewer's comments in this table. | Please see our point-by-point responses in this table. |
| | **Referee #2: Specific comments** | | |
| 2 | The authors highlight that NarCLIM2 has large improvements in tasmax biases, with small absolute biases of ~0.5K over many regions. Are these biases also evident when downscaling all individual GCMs, or simply in the ensemble mean? This relates to the order of operations of where the bias is computed (i.e. before or after the multi-model mean is computed). My concern is that there may be cancelling of biases (e.g. if one downscaled model has a warm bias and the other a cold bias). Can the authors confirm that this is not simply cancelling of biases? Related to this, showing biases for each downscaled model (perhaps in Supplementary material) would help to confirm this. | The reviewer is asking whether the ensemble mean is made from some models with positive bias and some models with negative bias so in the ensemble mean these biases somewhat cancel out. The answer is yes this is what happens with a reasonably good ensemble and indicates that the observations fall within the spread of the ensemble.

Results for individual NARCliM models are now provided in the revised manuscript. The overall magnitude of the individual biases within the ensemble were smaller in N2.0 compared to N1.x -- though there were some exceptions to that for some N2.0 individual | We had stated that NARCliM2.0 shows significant improvement in tasmax biases, primarily based on the ensemble mean. However, this improvement is also evident in several of the individual simulations, though there are exceptions. In NARCliM1.0 and 1.5, most simulations exhibited strong systematic cold biases. In contrast, for several ensemble members, NARCliM2.0 reduces these cold biases or replaces them with small warm biases. Overall, individual simulations in NARCliM2.0 generally show a reduction in bias compared to those in NARCliM1.0 and 1.5. |

| | | models -- please see revised text in column right now included in the revised manuscript. | This is shown for the individual simulations in the Supporting Information Figures S4-S6 for tasmax. Equivalent plots for tasmin (for which NARCliM 2.0 does not show improved performance versus NARCliM 1.x) are shown in Figures S8-S10, and for precipitation in Figures S12-14.

To make this clearer in the revised manuscript, we have revised the relevant section of text in Sect. 6.1 in the revised manuscript, and we now state the range of per-RCM biases for each variable in the revised main text. We also highlight RCMs that are in some way exceptions e.g.: (lines 648-662)

"Overall, NARCliM 2.0 RCMs simulate maximum temperature more accurately than NARCliM1.x, with widespread, statistically significant reductions in cold biases in the ensemble mean (Figure 9), as well as for many individual RCMs (Supporting Information Figure S4-S6). These reductions in bias apply for all timescales but are largest for the annual mean, i.e., the area-averaged mean absolute bias for the NARCliM 2.0 ensemble is 0.75K (range: 0.61 to 2.03 K), 1.73 K (range: 1.1 to 2.37 K) for NARCliM 1.5, and 1.89 K (range: 0.55 to 4.12 K) for NARCliM 1.0 (Figure 9d,g,j and Figure S4). Notably, the NARCliM2.0 ensemble mean annual mean maximum temperature bias magnitudes are small, i.e., around <0.5 K, over south-west WA, southern coastal regions, and several eastern regions. This may be important from a climate change adaptation and mitigation perspective as these |

| | | | regions are heavily populated and economically significant. NARCliM 2.0 retains warm biases of similar magnitude to NARCliM 1.5 along the north-west coast of Australia (Figure 9d,g). Moreover, these warm biases cover additional areas for NARCliM 2.0, especially during DJF (Figure 9e,h). A wide range of bias signs are evident for the individual NARCliM 2.0 ensemble members (Figures S4-S6) and a minority of NARCliM 2.0 RCMs retain strong cold biases, i.e. at an annual time NARCliM 2.0-NorESM2-MM R3 (mean absolute bias = 2.03 K) and UKESM-1-0-LL R3 (1.77 K)." |
|---|---|---|---|
| 3 | Some discussion of observational uncertainty seems warranted, especially if model biases are truly approaching 0.5K. | We agree, it is a good point to raise. The discussion in the panel right is now included on observational uncertainty, which is added at the end of section 4.1 in the revised manuscript. | Revised manuscript now states the text shown below (added to the Discussion, lines 909-923). "Consideration of observational uncertainty is warranted. We have evaluated NARCliM RCM skill via comparison with AGCD observations. Whilst AGCD are a high quality gridded observational data set, like any set of observations, they contain errors and uncertainties. Consequently, the outcomes of our evaluations depend on both the models being evaluated and the AGCD observational dataset. This is clearly a broader issue that applies to any model evaluation versus observations. Uncertainties in AGCD for temperature and precipitation arise from sparse station coverage in some locations, especially in remote areas, and interpolation errors in generating gridded data. More specifically, temperature uncertainties include urban heat island effects, inhomogeneities in observation records, and |

| | | | elevation differences. Precipitation uncertainties involve underestimation of extremes, rain gauge measurement errors, and challenges in representing complex terrain.

For our purposes, the question of how much of a bias of ~0.5 K is due to the model errors versus the observational uncertainty cannot be currently quantified, because the models are evaluated against this single observational dataset. This leaves the observational uncertainty as implicitly included in our results. In the future observational uncertainty could be explicitly considered using a method like the Observation Range Adjusted (ORA) statistics (Evans and Imran, 2024)." |
|---|---|---|---|
| 4 | The text and figures swap between K and Celsius units, best to choose one. | Thanks for pointing this out. We have made changes in the text and to the figures to keep the unit consistent as K throughout. | Temperature units are now K throughout the revised manuscript. |
| 5 | Obviously a large effort has gone into producing the convection-permitting resolution model output. However, the improvements are mostly seen in temperature and not in precipitation. Perhaps this is because the focus here is on evaluating mean precipitation and not extremes? Can the authors comment further on this? Referring and discussing other international literature here would be useful also. | In this study, the scope was to focus on an initial 'first-order' evaluation of mean precipitation rather than extremes of precipitation. However, clearly much valuable research can now be undertaken into evaluating the skill of NARCliM2.0 in simulating extreme precipitation, subdaily precipitation, etc, using NARCliM 2.0 20 km and 4 km data, especially since these data are now publicly available. A great avenue for further research is to assess the potential value-add in simulating extreme and subdaily precipitation at convection permitting scale versus the convection-parameterised 20 km data. This is now stated in the revised manuscript. | Text added to the revised manuscript as per column left / shown below (lines 889-897).

"More generally, the scope of the present study was to focus on an initial "first-order" evaluation of mean precipitation rather than extremes of precipitation. However, clearly valuable research can now be undertaken into evaluating the skill of NARCliM 2.0 in simulating extreme precipitation, subdaily precipitation, etc, using NARCliM 2.0 20 km and 4 km data, noting these data are now publicly available. A good avenue for further research is to assess the potential added value in simulating extreme and subdaily precipitation at convection permitting scale versus the convection-parameterised 20 km data. |

| | | In term of previous works: multiple studies have confirmed that convection-permitting resolution model can improve simulating daily and sub-daily rainfall extremes (Xie et al., 2024; Cannon and Innocenti, 2019; Kendon et al., 2017). In future work, we will also assess added value of convection-permitting resolution model in simulating precipitation related extremes.

Xie, K., Li, L., Chen, H., Mayer, S., Dobler, A., Xu, C.-Y., and Gokturk, O. M.: Enhanced Evaluation of Sub-daily and Daily Extreme Precipitation in Norway from Convection-Permitting Models at Regional and Local Scales, Hydrol. Earth Syst. Sci. Discuss. [preprint], https://doi.org/10.5194/hess-2024-68, in review, 2024.

Cannon, A. J. and Innocenti, S.: Projected intensification of sub-daily and daily rainfall extremes in convection-permitting climate model simulations over North America: implications for future intensity–duration–frequency curves, Nat. Hazards Earth Syst. Sci., 19, 421–440, https://doi.org/10.5194/nhess-19-421-2019, 2019.

Kendon, E. J., and Coauthors, 2017: Do Convection-Permitting Regional Climate Models Improve Projections of Future Precipitation Change?. Bull. Amer. Meteor. Soc., 98, 79–93, | Several previous studies have confirmed that convection-permitting resolution models can improve the simulation of daily and sub-daily rainfall extremes (Xie et al., 2024; Cannon and Innocenti, 2019; Kendon et al., 2017)." |

| | | https://doi.org/10.1175/BAMS-D-15-0004.1. | |
|---|---|---|---|
| 6 | On statistical significance. My personal view is that statistical significance is generally misunderstood and misinterpreted in climate science. However, I do think using significance in terms of model agreement is much more defensible (as you have done on top of this). If statistical significance is used, the authors also need to account for multiple testing (e.g. via the false discovery rate), which does not appear to be done:

https://journals.ametsoc.org/view/journals/bams/97/12/bams-d-15-00267.1.xml?tab_body=abstract-display | Thank you for your suggestion and the reference you have posted is interesting and something we have applied in the revised version of this manuscript and will continue to apply going forwards.

The ensemble mean based plots (Figures 9-14 and panels a, l and s in Figure 15) are the only plots where we combine multiple collections of null hypotheses. For these Figures 9-14 (and panels a, l, and s in figure 15) we have included revised plots with a corrected criterion using Walker's test using Eq.2 from the reference you provided. We applied Walker's test as this is stricter than FDR and easier to implement at this stage. Using this revised method, dependent on the NARCliM ensemble in question, alpha values change from 0.05 to alpha = 0.0051162 (for example). We found no major visible changes to the significance results / significance stippling of our plots for temperature biases and future projections, as can be observed in the comparison of original versus revised figure versions shown below this table. Here, the results are similar between the original version and the revised version implementing your suggestion, e.g. temperature climate change signals show widespread significant future changes. | Reviewer's suggestion implemented and Figures 9-15 revised in the revised manuscript as described in column left (please see also example figures below this table, pp. 21-23). Evaluation methods in the revised main text now states the additional text below (lines 212-213); results/figures in question revised throughout as indicated in column left:

"Significance thresholds were adjusted to account for multiple testing using Walker's test (Eq.2 in Wilks, 2016)". |

| | | Before implementing the reviewer's suggestion, the original results for precipitation climate change signals tended to be non-significant over most regions for most models. Having implemented the reviewer's suggestion, there are fewer locations showing statistically significant future changes for mean precipitation (see comparison figures below this table, pp. 21-23). | |
|---|---|---|---|
| 7 | In Figure 15, is there an understanding of why the projections for ACCESS-ESM1-5 projections are so dry? Presumably this is in the GCM also? Do we know why that is from the physical perspective? | ACCESS-ESM1-5 driven RCM simulations project very dry futures for Australia, which is mostly inherited from the GCM. There are 40 realisations for ACCESS-ESM1-5, but only realisation 6 provides sub-daily outputs that can be used in dynamical downscaling using WRF. This realisation simulates a particularly dry projection over Australia, especially for eastern Australia, making it a useful "stress test" case.  It also shows that internal variability within the GCM is a factor in producing this dry projection. Please see more details in: https://research.csiro.au/access/model-ensembles-to-understand-climate-variability-and-change/

In terms of GCM skill versus observations, globally, this GCM is dry biased over a few regions owing to a location bias with the Inter-tropical Convergence Zone (ITCZ), e.g. see Ziehn et al. (2020): CSIRO PUBLISHING | Journal of Southern Hemisphere Earth Systems Science | Text shown below added to the revised manuscript (lines 941-950).

"Some NARCliM 2.0 RCMs produce very similar precipitation projections for certain GCM-RCM combinations. Notably, ACCESS-ESM-1-5-R3 and R5 under SSP3-7.0 both produce widespread dry projections that are substantially drier than other NARCliM 2.0 models. This GCM projects very dry futures across Australia (Di Virgilio et al., 2022), so this result in the R3 and R5 RCMs could be largely inherited from the driving data. There are 40 realisations for ACCESS-ESM1-5, but only realisation 6 provides sub-daily outputs that can be used in dynamical downscaling using WRF. This realisation simulates a particularly dry projection over Australia, especially for eastern Australia, making it a useful "stress test" case. In terms of GCM skill versus observations, globally, this GCM is dry biased over a few regions owing to a location bias with the Inter-tropical Convergence Zone (Rashid et al., 2022; Ziehn et al., 2020)." |

| | | | |
|---|---|---|---|
| | | and:

Rashid et al. (2022): https://www.publish.csiro.au/es/fulltext/es21028 | |
| 8 | Table 1 is very helpful. Can an extra row on computational resources (core hours) be added? This would help emphasise how much more of an effort going to 4km resolution is. | Good suggestion. For NARCliM 2.0, during production phase of running both the 20 km and convection-permitting 4 km simulations, we used approximately **1060M core hours**. Note that these domains were run simultaneously, we do not have separate usage for the 4km resolution domain only.

For NARCliM1.5, figures used are from when we were performing cost estimates for NARCliM 2.0 estimates (i.e. not actual logs): we consumed in total **30M core hours**. Unfortunately, NCI (the HPC facility we used) discarded historical SU usage when they replace their main HPC, so we can not confirm the original billing logs.

Records for core hour usage for the original NARCliM 1.0 are unfortunately no longer available, but core hour usage per ensemble member year should be broadly similar to NARCliM 1.5. | Table 1 is updated accordingly and with an additional row in the revised manuscript (line 137). |
| 9 | Figure 4, for precip, are the units mm/day? | Thanks for asking this question, yes, the units are mm/day – figure caption revised accordingly. | Figure caption revised for units. |
| 10 | Figure 9 (and others), I found it difficult to see the stippling/hatching. The resolution of the file was low (not sure if this was an issue with the pdf | We agree that Figure 15 is difficult to read, e.g. the original version was 300 DPI; we have now increased the DPI to 600, among other | Figure 15 revised as suggested (please see example below this table, p.24). |

| | | | |
|---|---|---|---|
| | preprint?) but please ensure that high resolution figures are used and that the journal isn't compressing these in the final version. The resolution is particularly low for Figure 15 and very difficult to read. | modifications. We have revised this plot, please see the example new Figure 15 below this table (p. 24). | |
| 11 | I think in some figures there is a lot more repetitive text than there needs to be. Rethinking the layout headers of certain figures would help. For example, Figure 9 and 12, (Annual, DJF, JJA) could simply be headers at the top of each page, and the different versions of Narclim could be along the LHS of page. The text is often also too small to read. E.g. the colorbar caption in Figure 15 is excessively long and this information could simply be in the caption. | Thanks for these suggestions. We have revised these figures as you have suggested – please see examples below this table (pp. 21-23). | Figures modified as suggested in the revised manuscript (please see examples below this table, pp. 21-23). |

[Figure]

**Reviewer 2, Comments #6 and #11**. Left: original **Figure 9** from initial submission; Right: revised **Figure 9** using revised statistical significance method (please see #6) and revised plot layout/headers and labelling and increased DPI (please see #11)

[Figure]

**Reviewer 2, Comments #6 and #11**. Left: original **Figure 12** from initial submission; Right: revised **Figure 12** using revised statistical significance method (please see #6) and revised plot layout/headers and labelling and increased DPI (please see #11)

[Figure]

**Reviewer 2, Comments #6 and #11**. Left: original **Figure 14** from initial submission; Right: revised **Figure 14** using revised statistical significance method (please see #6) and revised plot layout/headers and labelling and increased DPI (please see #11)

[Figure]

**Figure 15**: revised version (Reviewer 2, comment #10)

**Table 3. Anonymous Referee 3 (RC3) Comments**

| # | Issue Description | Discussion | Revision (in re-submitted manuscript) |
|---|---|---|---|
| | **Referee #3: General Comments** | | |
| 1 | The authors present the regional climate model NARCliM2.0 and evaluate it using various GCM and RCM ensembles, as well as its precursor versions 1.0 and 1.x. The research topic is highly interesting, and the research work has been conducted meticulously and comprehensively, making it very valuable for regional climate model evaluation and future climate projections in Australia. The research framework is also inspiring for regional climate science, particularly for other regions with large populations. The manuscript is well-written and well-structured. In conclusion, I recommend publication in GMD after the specific comments listed below have been addressed. | We thank the reviewer for reviewing this manuscript, for the positive and constructive remarks on our work, and for recommending publication after addressing your specific comments below. | |
| | **Referee #3: Specific comments** | | |
| 2 | Line 81: "and 3) summarise the climate projections produced by CMIP6-NARCliM2.0 and how these" to "3) summarise the climate projections produced by CMIP6-NARCliM2.0 and how these". | Thanks for pointing this out – text changed as suggested. | Text revised (lines 81-82). |
| 3 | Line 83-88: "section x." to "Section x". Please check all "section x.x" and "sect. x.x" in the manuscript. | Agreed. | Text revised throughout as suggested. |
| 4 | Line 108-109: "NARCliM2.0 RCMs have a 20 km resolution CORDEX-Australasia domain (versus 50 km) and 4 km (versus 10 km) domain over southeast Australia and use 45 (versus 30) vertical levels". The horizontal resolution in NARCliM2.0 has more than doubled resolutions, yet the vertical resolution is from 30 to 45 vertical levels. What do authors think of the choice of 45 levels instead 60 or even more? | There is no strict requirement for vertical resolution to match horizontal resolution. However, in NARCliM 2.0, we carefully balanced the horizontal and vertical resolutions. By increasing the number of vertical levels from 30 to 45, we primarily enhanced the vertical resolution within the boundary layer, allowing for a better representation of | No change in the main text. |

| | | vertical profiles of temperature, moisture, and winds. The vertical grid spacing in the boundary layer is around 50–200 meters, which is sufficient to resolve important vertical processes. In early testing for NARCliM2.0, we also tested using 60 and 75 vertical levels. The surface climate produced was very similar to when using 45 levels, but the computational cost was substantially larger. Given that finding and resource constraints, we determined that 45 vertical levels could effectively meet the objectives of NARCliM 2.0. | |
|---|---|---|---|
| 5 | Line 142: "manuscripts describe elements shown in Figure 2, and which are therefore only summarised briefly in", remove "and". | Agreed: 'and' not needed. | Text revised as suggested (line 227) |
| 6 | Line 164-167: "The performances of the different test RCM configurations are evaluated, ultimately selecting a subset of seven RCMs for subsequent downscaling of ERA5 reanalysis and comprising the CORDEX evaluation experiment." To "The performance of the different test RCM configurations is evaluated, ultimately leading to the selection of a subset of seven RCMs for subsequent downscaling of ERA5 reanalysis as part of the CORDEX evaluation experiment". | Agreed, text revised as suggested. | Text revised (line 250). |
| 7 | Line 170: 'production' should be "production". Please check all 'something' in the manuscript. | In the revised manuscript, we have avoided the use of text like 'production' – production is sufficient, hence quotes removed as per example right. | Text revised as follows (line 255-256):

"Evaluating these ERA5-forced simulations informs selection of two definitive, production RCMs for CMIP6-forced downscaling" |
| 8 | Line 190-191: "Non-normally distributed variables (e.g. snow depth and precipitation) are checked for global minima and maxima only." To "Non-normally distributed variables (e.g., | Text in sentence corrected as suggested, including correction for all "e.g." as suggested. | Text revised throughout as suggested. |

| | | | |
|---|---|---|---|
| | snow depth and precipitation) are checked only for global minima and maxima." Please check all "e.g." in the manuscript. | | |
| 9 | Line 201: "Check that changes over time are within realistic ranges (i.e. assess temporal gradients)." To "Check that changes over time are within realistic ranges (i.e., assess temporal gradients)." Please check all "i.e." in the manuscript. | Text changed as suggested. | Text revised throughout. |
| 10 | Line 354-355: "Some studies have shown using this option improves modelling of soil moisture (e.g. Zhuo et al., 2019)." to "Some studies have shown **that** using this option improves **the** modeling of soil moisture (**e.g.,** Zhuo et al., 2019)." | Thanks – changes implemented as suggested. | Manuscript text revised (lines 453-454). |
| 11 | Table 9: I am confused about how exactly the "R1-R7" RCMs are shortlisted. It said in Line 609 that "RCMs are shortlisted from the set of 20 if they rank highly for both performance and independence", but it is not clear how the RCMs are ranked from "R1" to "R7". Please explain it in more detail. | We shortlisted the 7 RCMs from the shortlisted 20 candidates based on their performance and independence ranking. However, there was no ranking from R1 to R7 per se, this is just a naming convention chosen at the point of embarking on the next stage of the design/model evaluation process which was the ERA5-forced RCM simulations conducted for the CORDEX ERA5 evaluations. Only after completing these CORDEX ERA5 evaluations did we compare the performance of R1-R7 and at that point we selected R3 and R5 as the definitive, production RCMs for the subsequent CMIP6-forced RCM simulations – please see Di Virgilio et al.,([https://gmd.copernicus.org/preprints/gmd-2024-41/gmd-2024-41.pdf](https://gmd.copernicus.org/preprints/gmd-2024-41/gmd-2024-41.pdf)) for further details. | We have added the following note to the revised main text to provide clarification on this point (lines 630-632):

"We note here that R1-R7 are simply a chronological naming convention and do not imply any ranking for these 7 RCM configurations." |
| 12 | Figure 15: there are many subfigures and their titles are not easy to read. Please consider improve the visualization. | Agreed, the original Figure 15 was of insufficient quality (e.g. 300 DPI), so we | Figure 15 revised, please see example below this table. |

| | | have increased to 600 DPI and improved clarity of stippling and titles as far as is possible for a figure with 31 individual plot panels. | |
|---|---|---|---|
| 13 | Line 777: "with 4 RCMs using BMJ, 2 RCMs using Tiedtke, and 1 using Kain-Fritsch." Please give the references to the cumulus parameterisations. | This is a good idea – we have included references for all physics used in the study. | References for all physics settings used added into Table 3 (pp. 16-17) via an additional column in the revised manuscript. |

[Figure]

**Figure 15**: revised version